# DDM1 and Lsh remodelers allow methylation of DNA wrapped in nucleosomes

David B Lyons[1], Daniel Zilberman[1,2]*

[1]Department of Plant and Microbial Biology, University of California, Berkeley, Berkeley, United States; [2]Department of Cell and Developmental Biology, John Innes Centre, Norwich, United Kingdom

**Abstract** Cytosine methylation regulates essential genome functions across eukaryotes, but the fundamental question of whether nucleosomal or naked DNA is the preferred substrate of plant and animal methyltransferases remains unresolved. Here, we show that genetic inactivation of a single DDM1/Lsh family nucleosome remodeler biases methylation toward inter-nucleosomal linker DNA in *Arabidopsis thaliana* and mouse. We find that DDM1 enables methylation of DNA bound to the nucleosome, suggesting that nucleosome-free DNA is the preferred substrate of eukaryotic methyltransferases in vivo. Furthermore, we show that simultaneous mutation of DDM1 and linker histone H1 in *Arabidopsis* reproduces the strong linker-specific methylation patterns of species that diverged from flowering plants and animals over a billion years ago. Our results indicate that in the absence of remodeling, nucleosomes are strong barriers to DNA methyltransferases. Linker-specific methylation can evolve simply by breaking the connection between nucleosome remodeling and DNA methylation.

DOI: https://doi.org/10.7554/eLife.30674.001

*For correspondence:
daniel.zilberman@jic.ac.uk

## Introduction

Cytosine methylation provides a mechanism to heritably alter the genome without permanent modification of the DNA sequence (*Kim and Zilberman, 2014*; *Du et al., 2015*). In eukaryotes, Dnmt1 family methyltransferases, called MET1 in plants, rely on selective recognition of hemi-methylated symmetrical CG dinucleotides by an obligate cofactor (*Kim and Zilberman, 2014*). This mechanism semiconservatively propagates methylation patterns following cell division, and – especially in plants – across generations. Plant and animal genomes also contain cytosine methylation outside CG dinucleotides (*Du et al., 2015*). In plants, the CMT3 methyltransferase family catalyzes methylation of CNG trinucleotides, conventionally described as CHG (where H is any non-G base) to avoid overlapping CG sites (*Law and Jacobsen, 2010*). The related CMT2 family can methylate cytosines outside CG and CNG contexts (*Zemach et al., 2013*; *Stroud et al., 2014*), a pattern of specificity referred to as CHH. CMT2 and CMT3 preferentially methylate heterochromatic transposable elements (TEs) and rely on a positive feedback loop with dimethylation of lysine 9 of histone H3. Plants also possess an RNA-directed DNA methylation (RdDM) pathway, in which 24 nucleotide RNA molecules guide DRM methyltransferases (homologs of animal Dnmt3) to initiate DNA methylation in all sequence contexts, and to maintain CHH methylation at relatively euchromatic TEs (*Matzke and Mosher, 2014*).

As implied by the above description of methylation pathways, DNA methylation represses TE transcription and transposition in plants and vertebrates (*Law and Jacobsen, 2010*). Methylation also regulates endogenous genes: methylation close to the transcriptional start site can cause gene silencing, whereas methylation of other genic regions can promote or counteract expression

**eLife digest** Living cells add chemical tags to their DNA to regulate which genes are switched on or off at any given time. These tags include methyl groups added to one of the letters of the DNA code called cytosine. Both plants and mammals need cytosine methylation to develop properly. This methylation also keeps sections of foreign DNA that may have invaded the cell in check.

DNA inside the cell is tightly packed, wrapped around proteins to form spool-like structures called nucleosomes. Between each nucleosome is a short DNA segment called a linker region. The DNA wound into nucleosomes is generally inaccessible to other proteins, such as those that add methyl groups. Yet, in flowering plants and mammals, cytosine methylation occurs in both nucleosomes and in linker regions. It was not clear how DNA could be modified in the restrained setting of nucleosomes.

Enzymes called nucleosome remodelers can loosen nucleosomes to allow other proteins to reach the DNA. Lyons and Zilberman asked whether cytosine methylation occurs on the nucleosome-bound DNA or if it requires enzymes like these to free the DNA from the constraints of the nucleosome. The experiments involved a plant called *Arabidopsis thaliana* and mouse cells grown in the laboratory. In mutant plants lacking a nucleosome remodeler called DDM1, cytosine methylation occurred in the linker regions but not in the nucleosomes. Mouse cells lacking the mouse version of DDM1 also showed less cytosine methylation in the nucleosomes.

These results suggest that nucleosomes are barriers to the enzymes that modify DNA. Nucleosome remodeling enzymes like DDM1 can overcome these obstacles to enable cytosine methylation of nucleosome-wrapped DNA.

These findings imply that cytosine methylation is more easily established and maintained on nucleosome-free DNA. Abnormal patterns of DNA methylation have been linked to medical conditions – such as neurological disorders and cancers – and to plant defects that hamper agriculture. A better understanding of the process may in the future lead to ways to correct problems with cytosine methylation in these different contexts.

DOI: https://doi.org/10.7554/eLife.30674.002

(*Jones, 2012*). Genetic defects in the methylation machinery lead to human diseases such as cancer (*Baylin and Jones, 2016*) and disruption of methylation patterns during clonal propagation of plants causes developmental defects that hamper agriculture (*Ong-Abdullah et al., 2015*; *Springer and Schmitz, 2017*). Faithful propagation of DNA methylation is clearly essential, yet our understanding of the underlying processes is far from complete. In particular, we do not know how DNA methyltransferases contend with nucleosomes, which comprise the basic repeating units of chromatin.

Nucleosomes consist of about 147 bp of DNA tightly wrapped around an octameric complex of histone proteins, with the major and minor grooves of the double helix alternately facing toward and away from the histones roughly every 10 bp (*Luger et al., 1997*). Nucleosomes are separated by short stretches of linker DNA, which are variably bound by histone H1 (*Raghuram et al., 2009*). Flowering plants and mammals show an overall enrichment of methylation over nucleosomes, suggesting that the nucleosome is the preferred target for DNA methylation (*Chodavarapu et al., 2010*; *Chen et al., 2015*). Furthermore, a mild but robust 10 bp DNA methylation periodicity in both lineages suggests that methyltransferases directly modify DNA that is wrapped in nucleosomes (*Chodavarapu et al., 2010*). However, mammalian methyltransferase activity is inhibited by nucleosomes and H1 in vitro (*Robertson et al., 2004*; *Takeshima et al., 2006*; *Takeshima et al., 2008*; *Felle et al., 2011*; *Schrader et al., 2015*), and regions of the human and mouse genomes with reliably positioned nucleosome arrays organized by CTCF binding exhibit preferential methylation of linker DNA (*Kelly et al., 2012*; *Baubec et al., 2015*). Methylation is also almost completely confined to linker DNA in several diverse species of marine algae (*Huff and Zilberman, 2014*), suggesting that nucleosomes are generally refractory to DNA methylation. The reasons for the apparent discrepancy between in vitro and in vivo methyltransferase preferences, and for the observed interspecies differences, remain unknown.

Adding to the mystery, plants and mammals depend on the DDM1/Lsh family of Snf2-class nucleosome remodelers – proteins that can alter nucleosomes in various ways – to achieve normal levels of DNA methylation (*Jeddeloh et al., 1999*; *Zhou et al., 2016*). *Arabidopsis* plants with mutations in *DDM1* suffer major methylation losses in all sequence contexts, primarily in heterochromatic TEs, though much smaller losses are also observed in genes (*Jeddeloh et al., 1999*; *Teixeira et al., 2009*; *Zemach et al., 2013*; *Ito et al., 2015*). Knockout of mouse Lsh likewise causes major depletion of DNA methylation from repetitive heterochromatin (*Dennis et al., 2001*), with more variable effects on methylation in genes and other sequences (*Myant et al., 2011*; *Tao et al., 2011*). Genetic inactivation of *Arabidopsis* histone H1 partially rescues the *ddm1* hypomethylation phenotype, suggesting that DDM1 facilitates methyltransferase access to H1-containing chromatin (*Zemach et al., 2013*). However, how DDM1 does this is unknown, and the incomplete rescue of DNA methylation in *h1ddm1* compound mutants remains unexplained. More generally, how nucleosome remodelers facilitate methylation in species in which methyltransferases are proposed to act on the nucleosome surface is unclear.

Here we explore how DNA methylation is regulated by DDM1/Lsh, H1 and nucleosomes. We find that nucleosomes and H1 are barriers to DNA methylation and that DDM1/Lsh remodelers are required for methylation of nucleosomal DNA. At heterochromatic loci with reliably positioned arrays of nucleosomes, methylation in *h1ddm1* plants is strikingly periodic, with nearly wild-type levels in linkers and nearly *ddm1* levels in nucleosomes. Our results indicate that methyltransferases generally require remodeling activity to access nucleosomal DNA. Lack of such remodeling biases methylation toward linker DNA, providing a straightforward explanation for the linker-specific methylation patterns found in diverse eukaryotes.

## Results

### DDM1 enables CG and non-CG methylation of DNA wrapped in nucleosomes

To investigate the relationship between nucleosomes and DNA methylation, we deep sequenced unamplified DNA isolated from micrococcal nuclease (MNase)-digested chromatin of *Arabidopsis* plants with inactivating mutations in both canonical linker histone H1 genes (*h1*), *ddm1* mutants, compound *h1ddm1* mutants, and in WT controls. We selected mononucleosomes in silico to avoid inconsistencies stemming from DNA electrophoretic mobility. Using biological replicates, we defined four groups of nucleosomes per genotype, from well-positioned to poorly-positioned (see Materials and methods), and anchored methylation analysis at the presumptive dyad (nucleosome center) (*Chen et al., 2014*). We focused our initial analysis on heterochromatic sequences (see Materials and methods), where H1 is most abundant (*Rutowicz et al., 2015*) and DDM1 is most important for maintenance of DNA methylation (*Zemach et al., 2013*).

CG methylation shows a mild but clear depletion over well-positioned nucleosomes in WT and *h1* mutant plants, and an overt depletion in *ddm1* and *h1ddm1* plants (*Figure 1A*, *Figure 1—figure supplement 1*, loci with well-positioned nucleosomes on the left). Poorly positioned nucleosomes by definition do not have a well-delineated core region, and these loci accordingly exhibit a relatively flat level of methylation in all genotypes (*Figure 1A*, *Figure 1—figure supplement 1*). The difference between core and linker methylation is much greater in *h1ddm1* than in *ddm1* (*Figure 1B*), an effect that is observable at individual loci (*Figure 1C*, *Figure 1—figure supplement 2*), where methylation of linker regions – much stronger in *h1ddm1* than in *ddm1* – alternates with weak or absent methylation of nucleosome cores.

For the above analyses, we defined quality of nucleosome positioning separately for each genotype, and therefore the nucleosomes that comprise each of the four positioning groups vary between genotypes. To control for any potential sequence bias across genotypes, we directly compared the best positioned nucleosomes shared by WT and *h1ddm1*, or by *h1* and *h1ddm1* (*Figure 1—figure supplement 3*; see Materials and methods). This analysis produced the same pattern as in *Figure 1*, at nearly the same scale. Interestingly, using only WT well-positioned nucleosomes severely attenuates the core-to-linker CG methylation difference in *ddm1* and essentially eliminates any difference in *h1ddm1* (*Figure 1—figure supplement 3*), illustrating the importance of genotype-specific nucleosome mapping.

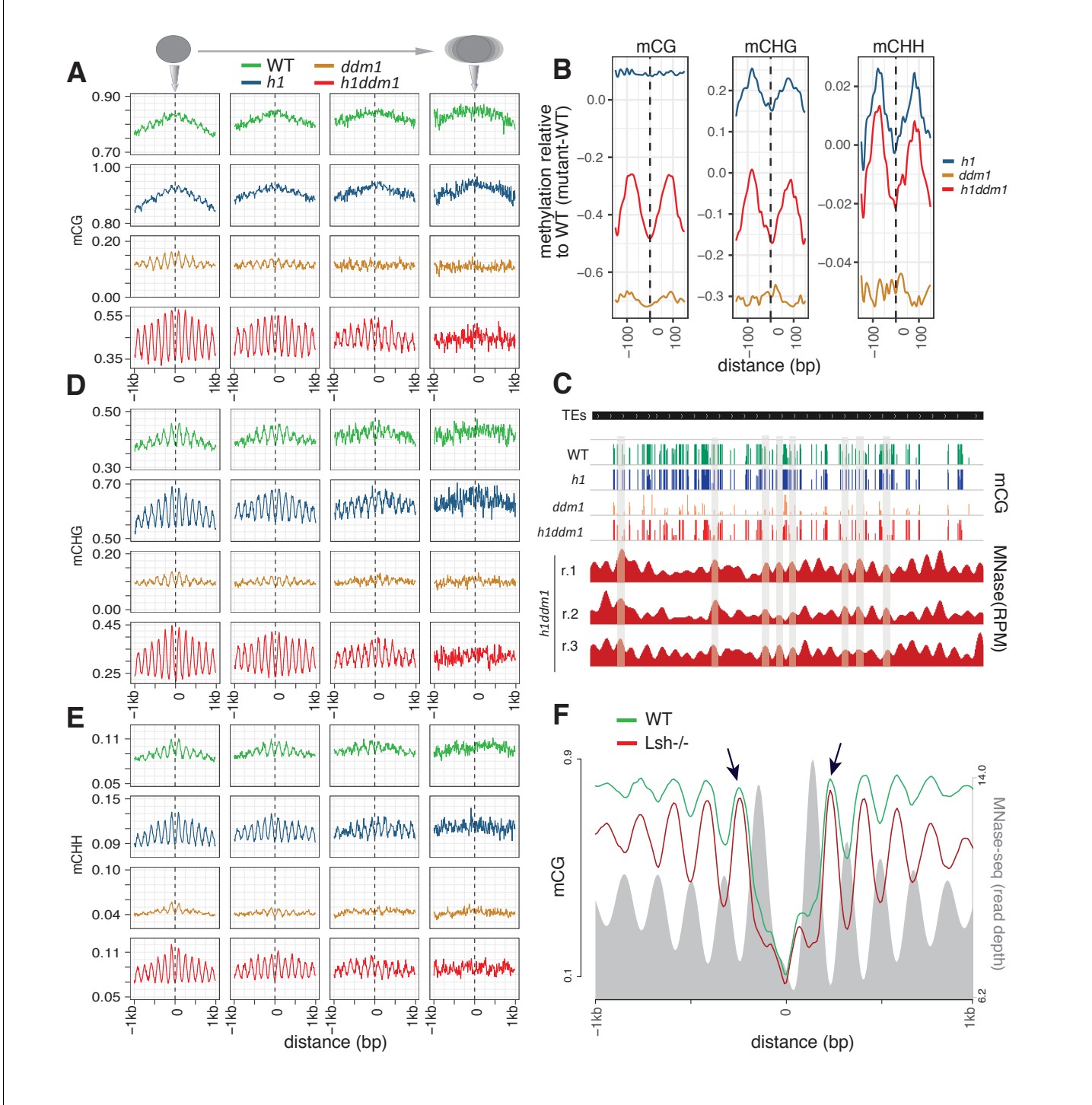

**Figure 1.** *Arabidopsis* and mouse nucleosomes are refractory to DNA methylation in the absence of DDM1/Lsh. (A) Average CG methylation (mCG) plotted around heterochromatic nucleosomes grouped by positioning reliability, with better-positioned nucleosomes on the left. (B) Average methylation plotted around well-positioned heterochromatic nucleosomes (leftmost group in panel A) with WT methylation subtracted from that of the indicated mutant. (C) Genome browser view at AT1TE53390 of mCG in the indicated genotypes, with smoothed *h1ddm1* MNase-seq shown in red (biological replicates shown as r.1–3). Vertical grey windows highlight well-positioned nucleosomes with core nucleosome depletion in *ddm1* and *h1ddm1*. Methylation scale is 0 to 1, MNase-seq scale is 0 to 0.25 reads per million (RPM). (D–E) Average CHG methylation (D) or CHH methylation (E) is plotted around nucleosomes, as in (A). (F) Lsh null and WT mCG from mouse embryonic fibroblasts is plotted around CTCF binding sites. Nucleosome profiles from the same cell type are depicted in grey. Arrows point to methylated linkers closest to CTCF sites. *Figure 1—figure supplement 1* presents the same analysis as *Figure 1A,D,E* but zoomed in around the anchoring nucleosome. *Figure 1—figure supplement 2* shows

*Figure 1 continued on next page*

*Figure 1 continued*

genomic regions of *Arabidopsis* exhibiting specific loss of nucleosome core methylation in *h1ddm1* plants. *Figure 1—figure supplement 3* illustrates that well positioned nucleosomes shared by *h1ddm1* and other genotypes exhibit the same methylation phenotype as shown in this figure.
DOI: https://doi.org/10.7554/eLife.30674.003
The following figure supplements are available for figure 1:

**Figure supplement 1.** DNA methylation around nucleosome cores.
DOI: https://doi.org/10.7554/eLife.30674.004
**Figure supplement 2.** Linker-biased CG methylation in *h1ddm1* plants.
DOI: https://doi.org/10.7554/eLife.30674.005
**Figure supplement 3.** Well-positioned nucleosomes shared across genotypes exhibit nucleosome core depletion and linker enrichment.
DOI: https://doi.org/10.7554/eLife.30674.006

Our results indicate that nucleosomes and H1 are obstacles to DNA methylation in vivo. Nucleosomes, which form much more stable associations with DNA than H1 (*Misteli et al., 2000*), are more powerful obstacles. In WT an *h1* plants, DDM1 activity allows DNA methyltransferases robust access to DNA. In *ddm1* mutants, nucleosomes are inaccessible and H1 blocks linker DNA, but the greater accessibility of linkers allows higher methylation levels at these sequences (*Figure 1A,B*). In the absence of DDM1 and H1, linker DNA is accessible to methyltransferases, but nucleosome cores are not, producing the robust methylation periodicity in *h1ddm1* plants (*Figure 1A–C*).

Interestingly, non-CG methylation shows a pronounced nucleosome core depletion at well-positioned nucleosomes in all genotypes (*Figure 1D,E*, *Figure 1—figure supplement 3*). In the CHG context, *h1ddm1* has the most exaggerated core-to-linker methylation differential, followed by *h1* (*Figure 1B,D*, *Figure 1—figure supplement 3*). In the CHH context, *h1ddm1* and *h1* show a similarly strong core-to-linker methylation differential (*Figure 1B,E*, *Figure 1—figure supplement 3*). Thus, whereas DDM1 allows for near-perfect efficiency of nucleosome core CG methylation (*Figure 1A*), non-CG methylation of nucleosomal DNA is well below that of linkers even in WT (*Figure 1D,E*). Loss of H1 increases the contrast between linker and core methylation levels (*Figure 1A–E*). Taken together, the observed CG and non-CG methylation patterns strongly argue that in the absence of nucleosome remodeling by DDM1, DNA within heterochromatic nucleosomes is unavailable to methyltransferases.

## Mouse Lsh facilitates methylation of nucleosomal DNA

DDM1 belongs to an ancient protein family that includes mouse Lsh, inactivation of which causes extensive DNA hypomethylation (*Dennis et al., 2001*; *Law and Jacobsen, 2010*). To assess whether Lsh, like DDM1, permits methylation of DNA wrapped in nucleosomes, we examined published methylation data from Lsh null mouse embryonic fibroblasts and WT controls (*Yu et al., 2014*) in relation to nucleosome positions (*Teif et al., 2012*) (*Figure 1F*). Because this analysis requires reliably well-positioned nucleosomes, we assessed regions of the mouse genome immediately flanking CTCF binding sites, which are surrounded by positioned nucleosome arrays (*Fu et al., 2008*). As previously reported, we find that WT DNA methylation is periodic around CTCF binding sites, with clear depletion at nucleosome core regions (*Kelly et al., 2012*; *Baubec et al., 2015*) (*Figure 1F*). Cells with inactivated Lsh show a much stronger methylation periodicity, with far greater depletion at nucleosome cores (*Figure 1F*). This contrast is most evident in the linkers that extend from the most reliably positioned nucleosomes on either side of the CTCF binding sites (emphasized with arrows in *Figure 1F*), which show nearly WT levels of DNA methylation in Lsh mutant cells. Our results indicate that nucleosomes present barriers to mammalian as well as plant DNA methyltransferases, and that the conserved function of the DDM1/Lsh family of remodelers is to enable methylation of nucleosomal DNA.

## Nucleosomes within euchromatic TEs impede DNA methylation

*Arabidopsis* TEs can be roughly grouped into heterochromatic and euchromatic elements (*Zemach et al., 2013*). Heterochromatic TEs, which we analyzed in *Figure 1*, tend to be longer, are methylated at CHH sites primarily by CMT2 and require DDM1 for methylation in all contexts (*Zemach et al., 2013*; *Stroud et al., 2014*). Euchromatic TEs are much shorter, less dependent on

DDM1, and CHH methylation at these elements is maintained by DRM1 and DRM2 (*Zemach et al., 2013*; *Stroud et al., 2014*). Because DRM1/2-dependent loci are short and often bordered by unmethylated DNA, analysis of methylation anchored to the centers of differentially methylated regions (DMRs; see Materials and methods) between *drm1drm2* mutants and wild-type produces a sharp peak (plots in the right column of *Figure 2A–C*). Nonetheless, anchoring methylation analysis to well-positioned nucleosomes shows that CG and CHG methylation at DRM-dependent loci is depleted over nucleosomes much as it is in heterochromatin (*Figure 2A–B*, *Figure 2—figure supplement 1*), demonstrating that nucleosomes in euchromatic TEs are refractory to DNA methylation. CHH methylation is also depleted over nucleosomes, but unlike in heterochromatin, the depletion is not obviously enhanced in any of the mutant genotypes (*Figure 2C*, *Figure 2—figure supplement 1*), probably because the RdDM pathway that guides DRM methyltransferases is largely independent of DDM1 but is associated with other remodelers (*Kanno et al., 2004*; *Smith et al., 2007*; *Zemach et al., 2013*; *Han et al., 2015*). As expected, CMT2-dependent loci recapitulate the behavior of heterochromatic TEs (*Figure 2D–F*, *Figure 2—figure supplement 1*). Due to the strong linker enrichment of methylation in *h1ddm1* heterochromatin (*Figure 1A–E*), even analyses centered on DMRs between *cmt2* and wild-type without regard to nucleosome position produce periodic *h1ddm1* methylation patterns (plots in the right column of *Figure 2D–E*).

## DDM1 facilitates DNA methylation of nucleosomes in genes

Previous work demonstrated a global enrichment of DNA methylation on *Arabidopsis* nucleosomes (*Chodavarapu et al., 2010*). We have so far described the opposite pattern, but our analysis has been confined to TEs. To identify regions of the genome that show the reported preferential methylation of nucleosomal DNA, we expanded our analysis to include all nucleosomes. We find that nucleosomes within genes possess a marked enrichment of CG methylation in WT, whereas TEs exhibit no depletion, so that methylation is enriched in WT nucleosomes on average (*Figure 3A*; see Materials and methods). Separating genic nucleosomes by degree of positioning, as we did for heterochromatin (*Figure 1A*), reveals an enrichment of DNA methylation in the nucleosome core that diminishes with increased positioning uncertainty (*Figure 3—figure supplement 1*). The correspondence between nucleosomes and DNA methylation within genes can be easily seen at individual loci (*Figure 3B*). Interestingly, nucleosomes in the exons of genes show a clear enrichment of DNA methylation, whereas intronic nucleosomes do not (*Figure 3C*), indicating that methylation is not intrinsically targeted to genic nucleosomes.

The enrichment of methylation over nucleosomes in genes does not preclude the possibility that this methylation requires a remodeling activity. The *ddm1* and *h1ddm1* mutations do not eliminate methylation of genic nucleosomes (*Figure 3A*, *Figure 3—figure supplement 1*), which is consistent with the preferential requirement of DDM1 for heterochromatic methylation (*Zemach et al., 2013*). However, because DDM1 does contribute to DNA methylation in genes (*Zemach et al., 2013*), we asked whether DDM1 is needed for methylation of a subset of genic nucleosomes. Indeed, we find that lowly-expressed genes depend on DDM1 to maintain nucleosome methylation, whereas more highly expressed genes do not, on average, need DDM1 (*Figure 3D*). Our definition of genes excludes TEs or sequences with TE-like methylation patterns (*Zemach et al., 2013*), therefore this result is not caused by TE contamination of our gene anotations. Genes with lower levels of expression undergo less transcription-coupled nucleosome displacement (*Workman, 2006*), and their methylation may therefore be more dependent on DDM1-mediated remodeling than more highly expressed counterparts. Our results demonstrate that DDM1 facilitates methylation of nucleosome-wrapped DNA in a subset of genes as well as in heterochromatin.

## The 10 bp DNA methylation periodicity requires DDM1/Lsh

Perhaps the most compelling evidence that DNA methylation occurs on the surface of the nucleosome is the reported subtle 10 bp periodicity of methylation in all sequence contexts (*Chodavarapu et al., 2010*) – a periodicity that reflects the alternate facing of the major and minor grooves of DNA toward and away from the histone octamer. If the 10 bp periodicity is indeed caused by differential accessibility of DNA on the nucleosome surface, it should not be affected by mutation of a nucleosome remodeler. To assess whether DDM1 is required for this pattern, we examined DNA methylation periodicity in *ddm1* plants and sibling controls at heterochromatic TEs.

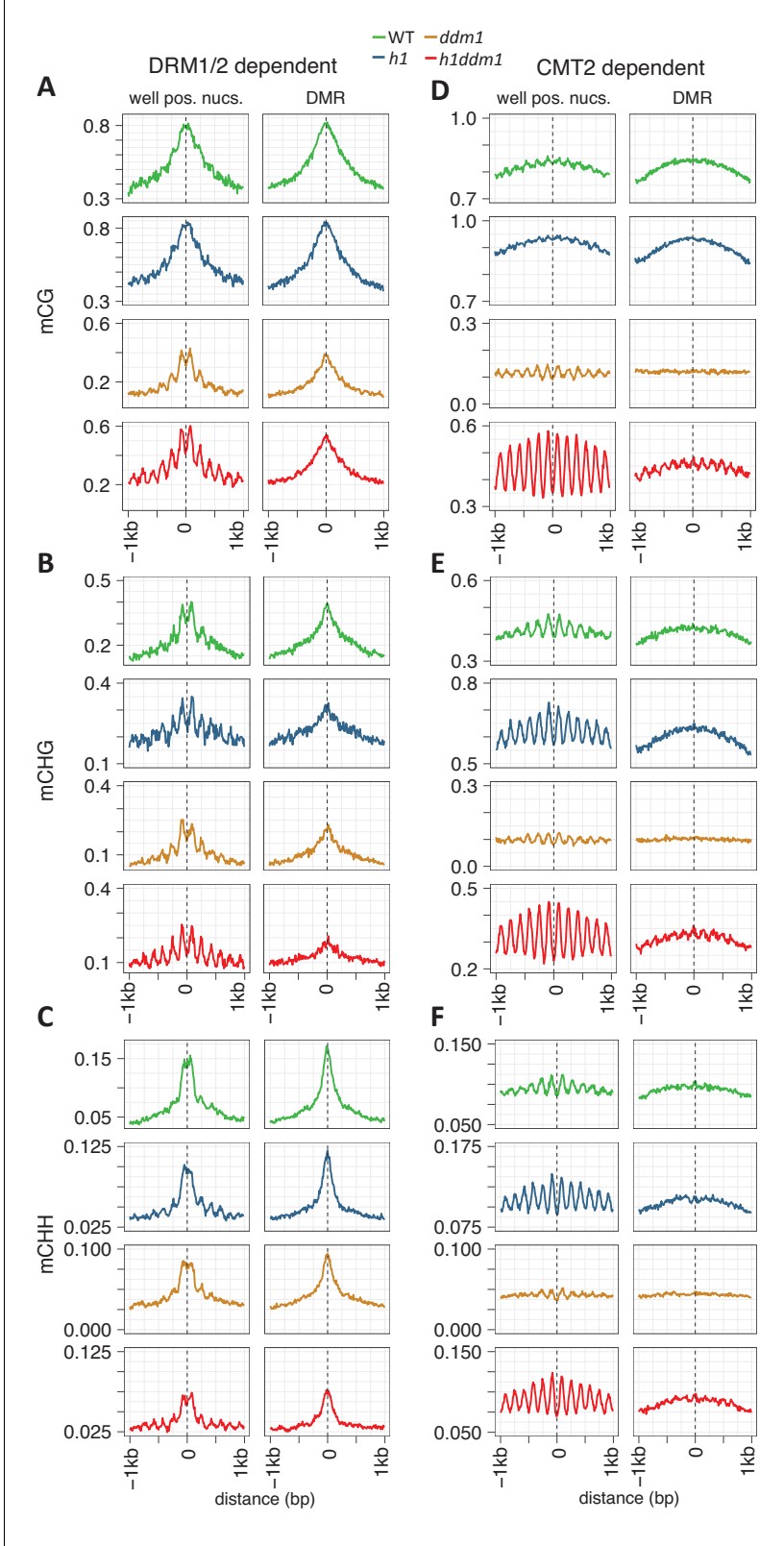

**Figure 2.** Nucleosomes within euchromatic TEs require DDM1 for DNA methylation. (**A–C**) DNA methylation in DRM1/2-dependent differentially methylated regions (DMRs) for the given sequence context anchored at the dyads of the nucleosomes in the best-positioned group (defined as in *Figure 1*; left panels) or anchored at the centers of DMRs lacking positioned nucleosomes (right panels). (**D–F**) CMT2 DMRs; otherwise as in (**A–C**) (see Materials and methods for full description and definition of DMRs). *Figure 2—figure supplement 1* shows the same data, but zoomed in on both axes.
*Figure 2 continued on next page*

*Figure 2 continued*

DOI: https://doi.org/10.7554/eLife.30674.007

The following figure supplement is available for figure 2:

**Figure supplement 1.** Zoomed-in view of DNA methylation around nucleosomes at DRM- or CMT2-dependent loci.

DOI: https://doi.org/10.7554/eLife.30674.008

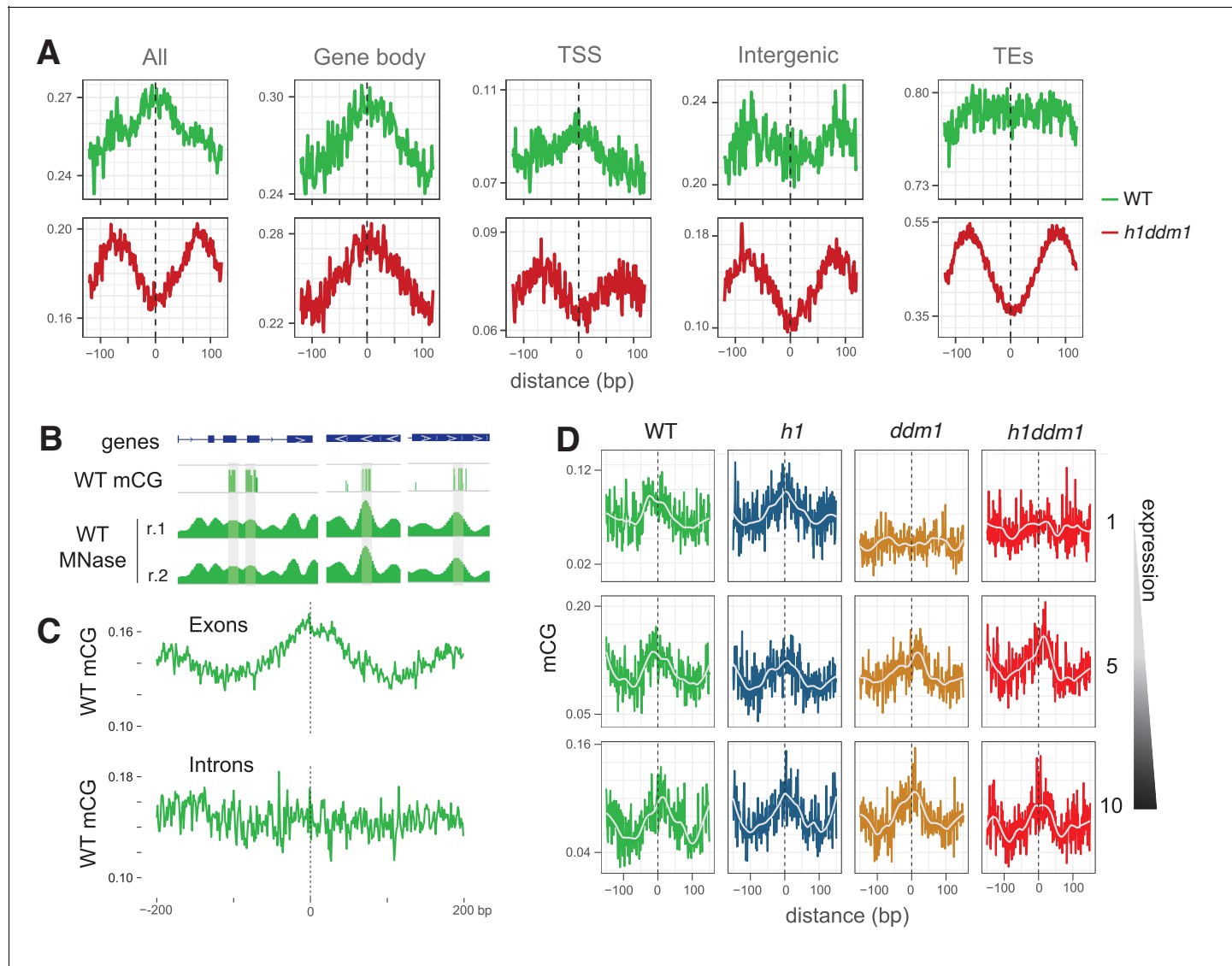

**Figure 3.** DDM1 facilitates the methylation of nucleosomes in genes. (**A**) Average CG methylation at all nucleosomes, irrespective of positioning, for the indicated genomic context. TSS, transcription start site; TEs, transposable elements. (**B**) Representative genome browser snapshots showing CG methylation (mCG) at genic nucleosome cores in WT. Methylation scale is 0 to 1, MNase-seq scale is 0 to 0.2 reads per million (RPM). (**C**) All nucleosomes in genes (TSS and gene body combined), irrespective of positioning, were characterized as either exonic (any overlap with exon) or intronic (only 100% overlap with intron). Average CG methylation is plotted centered on the nucleosomes in the two groups. (**D**) Average CG methylation at well-positioned nucleosomes in genes; lowest expression decile (1), middle decile (5), and the highest expression decile (10) are shown. *Figure 3—figure supplement 1* shows the enrichment of CG methylation on genic nucleosomes as a function of positioning reliability.

DOI: https://doi.org/10.7554/eLife.30674.009

The following figure supplement is available for figure 3:

**Figure supplement 1.** DNA methylation is enriched in genic nucleosomes.

DOI: https://doi.org/10.7554/eLife.30674.010

We anchored methylation analysis to the ends of 147 bp MNase fragments isolated in silico from our paired-end libraries, and calculated the strength of periodicity in the average per-base methylation signal with fast Fourier transform (FFT) computation (*Chodavarapu et al., 2010*; see Materials and methods).

As expected, WT plants display a 10 bp methylation periodicity in all sequence contexts (*Figure 4A*, *Figure 4—figure supplement 1A–C*). Lack of H1 does not appreciably alter the periodicity in non-CG contexts; CG methylation periodicity is weaker than WT (*Figure 4A*), probably because the very high levels of heterochromatic CG methylation in *h1* plants (*Zemach et al., 2013*) leave little room for oscillations. Importantly, the 10 bp periodicity is virtually absent in *ddm1* heterochromatin in all contexts (*Figure 4A*). In contrast, methylation of genic nucleosomes, which in most cases does not require DDM1, displays a clear 10 bp periodicity in all genotypes examined (*Figure 4B*, *Figure 4—figure supplement 1D*).

To confirm our *Arabidopsis* results, we examined whether the 10 bp periodicity also declines in Lsh mutants. Because our analysis of Lsh methylation is based on published nucleosome data from WT cells (*Teif et al., 2012*), and corresponding data from Lsh null cells is not available, we based our calculation of methylation periodicity on nucleosomal fragments mapping around the CTCF sites used in *Figure 1F*, where nucleosomes positioning is highly reliable (*Fu et al., 2008*). We find a clear 10 bp periodicity in WT CG methylation, but in Lsh-/- the signal is noisy and does not exhibit a maximum at 10 bp (*Figure 4C*, *Figure 4—figure supplement 2*). Although this phenotype may be caused by anchoring the Lsh-/- methylation analysis to WT rather than Lsh-/- nucleosomes, bound CTCF sites generally possess very well positioned flanking nucleosomes (*Fu et al., 2008*; *Stadler et al., 2011*), suggesting that methylation periodicity is attenuated in the absence of Lsh. Thus, our results indicate that the 10 bp methylation periodicity observed in *Arabidopsis* and mouse

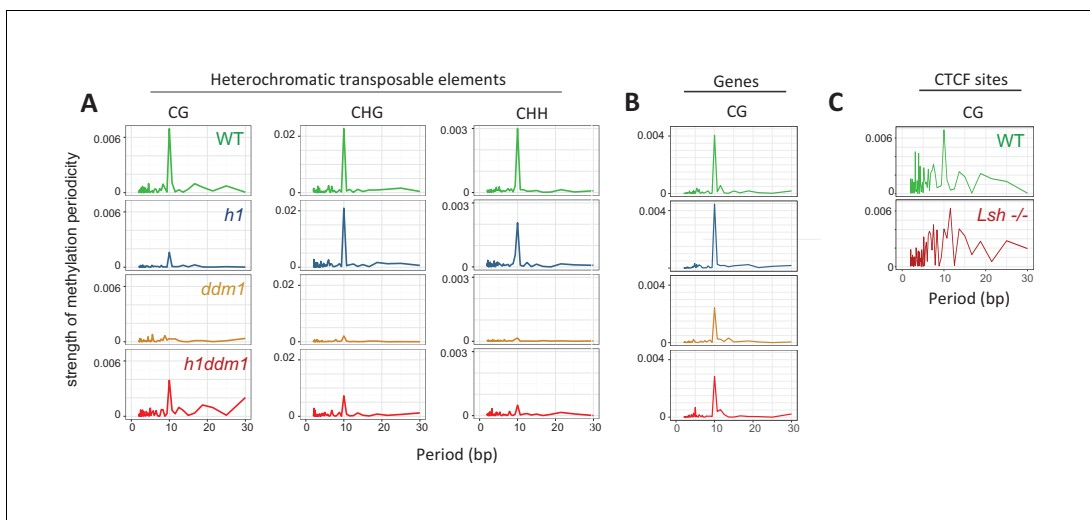

**Figure 4.** Methylation periodicity with respect to the histone octamer requires DDM1 and Lsh. (**A–B**) Fast Fourier transform (FFT) periodogram of average DNA methylation across all 147 bp MNase-seq fragments in heterochromatic TEs (**A**) and genes (**B**) in *Arabidopsis* for the indicated genotype. (**C**) Same as (**A–B**), but with 147 bp MNase-seq fragments only from mouse CTCF sites, as in *Figure 1F*. *Figure 4—figure supplement 1* shows the averaged *Arabidopsis* methylation data used for calculating periodicity by FFT while *Figure 4—figure supplement 2* shows the averaged mouse methylation data.
DOI: https://doi.org/10.7554/eLife.30674.011

The following figure supplements are available for figure 4:

**Figure supplement 1.** DDM1 is required for 10 bp methylation periodicity at heterochromatic nucleosomes.
DOI: https://doi.org/10.7554/eLife.30674.012

**Figure supplement 2.** WT and *Lsh*-/- average CG methylation at WT 147 bp MNase-seq fragments overlapping CTCF sites.
DOI: https://doi.org/10.7554/eLife.30674.013

requires a nucleosome remodeler, providing further support for the hypothesis that DNA methyltransferases cannot directly methylate nucleosomes.

## Arrays of positioned nucleosomes drive dramatic methylation phasing

The depletion of nucleosome methylation in *ddm1* and *h1ddm1* plants (*Figure 1A–E*, *Figure 1—figure supplement 3*) is reminiscent of the linker-specific DNA methylation patterns observed in three lineages of marine algae (*Huff and Zilberman, 2014*) that diverged from one another, and from flowering plants and animals, over a billion years ago (*Parfrey et al., 2011*). However, the contrast between the full methylation of linker DNA and the absence of nucleosomal methylation in marine algae (*Huff and Zilberman, 2014*) is much greater than that observed even in the best positioned group of *h1ddm1* nucleosomes (*Figure 1A*). A potential explanation is that the relevant algal species have exceptionally well-positioned nucleosomes (*Huff and Zilberman, 2014*), whereas even the best positioned *Arabidopsis* nucleosome group we identified is likely quite heterogeneous.

To test this hypothesis, we identified arrays of well-positioned *Arabidopsis* nucleosomes by using an unsupervised clustering algorithm (*Stempor et al., 2016*) to organize *h1ddm1* CG methylation data centered on heterochromatic nucleosomes in the best-positioned group. One of the resultant clusters (C2, 4849 loci) exhibits highly structured CG methylation that alternates with nucleosome positions (*Figure 5A*). All genotypes show methylation periodicity at C2 in all sequence contexts (*Figure 5A–D*). Remarkably, *h1ddm1* linker CG methylation reaches near-WT levels in this cluster, whereas nucleosome core methylation is close to *ddm1* levels (*Figure 5B*). The periodicity of *h1ddm1* non-CG methylation at C2 is also striking, with *ddm1*-like core methylation and linker methylation that is even higher than WT (*Figure 5C,D*, *Figure 5—figure supplement 1*). The *h1ddm1* methylation patterns at C2 closely resemble those seen in algae over a billion years removed from the *Arabidopsis* lineage.

## Discussion

We have shown that nucleosomes are strong impediments to DNA methylation and that the DDM1/Lsh remodeling proteins facilitate nucleosome methylation in vivo. It remains formally possible that DDM1/Lsh remodelers permit methyltransferases to work directly on the nucleosome surface or promote catalysis without affecting methyltransferase access. However, these interpretations are not parsimonious, and are inconsistent with the accumulating evidence that naked DNA is the preferred template for most DNA modifying enzymes. Such enzymes include cytidine deaminases (*Kodgire et al., 2012*), the TET1 dioxygenase (*Kizaki et al., 2016*), CAS9 (*Horlbeck et al., 2016*), and a bacterial cytosine methyltransferase that has the same catalytic mechanism as eukaryotic enzymes (*Kelly et al., 2012*). In conjunction with these observations, our data strongly support the hypothesis that cytosine methyltransferases generally require naked DNA, and that DDM1/Lsh nucleosome remodelers provide this substrate.

We propose that DDM1/Lsh-type remodelers render a region of nucleosomal DNA accessible to DNA methyltransferases in a manner analogous to the remodeling mechanism proposed for Swi2/Snf2 (*Zofall et al., 2006*). These remodelers, which are closely related to DDM1, bind the nucleosome and translocate DNA from linker toward dyad, generating a nucleosome-free DNA loop (*Figure 6*) (*Flaus et al., 2006*; *Zofall et al., 2006*; *Zhou et al., 2016*). Such a loop would allow methyltransferases access to the DNA that would be biased by its orientation to the histone octamer surface, plausibly producing the observed subtle 10 bp methylation periodicity. The 10 bp periodicity could also arise if the formation or stability of nucleosomes is affected by the orientation of 5-methylcytosine toward the histone octamer (*Jimenez-Useche et al., 2013*).

Despite the overall importance of DDM1 for DNA methylation, many nucleosomes, particularly those in genes, are methylated normally without DDM1 (*Figure 3*). A likely explanation is that other nucleosome remodelers facilitate methylation of euchromatic loci. This hypothesis is consistent with the observations that the RdDM pathway requires remodelers other than DDM1 (*Kanno et al., 2004*; *Smith et al., 2007*; *Zemach et al., 2013*; *Han et al., 2015*), although their specific functions remain unknown. The partial restoration of the 10 bp methylation periodicity in *h1ddm1* heterochromatin (*Figure 4A*) also suggests that remodelers other than DDM1 can mediate methylation of heterochromatic nucleosomes in the absence of H1. This idea is supported by the reported ability of H1 to prevent nucleosome remodeling in vitro for some Swi2/Snf2-type remodelers (*Lusser et al.,*

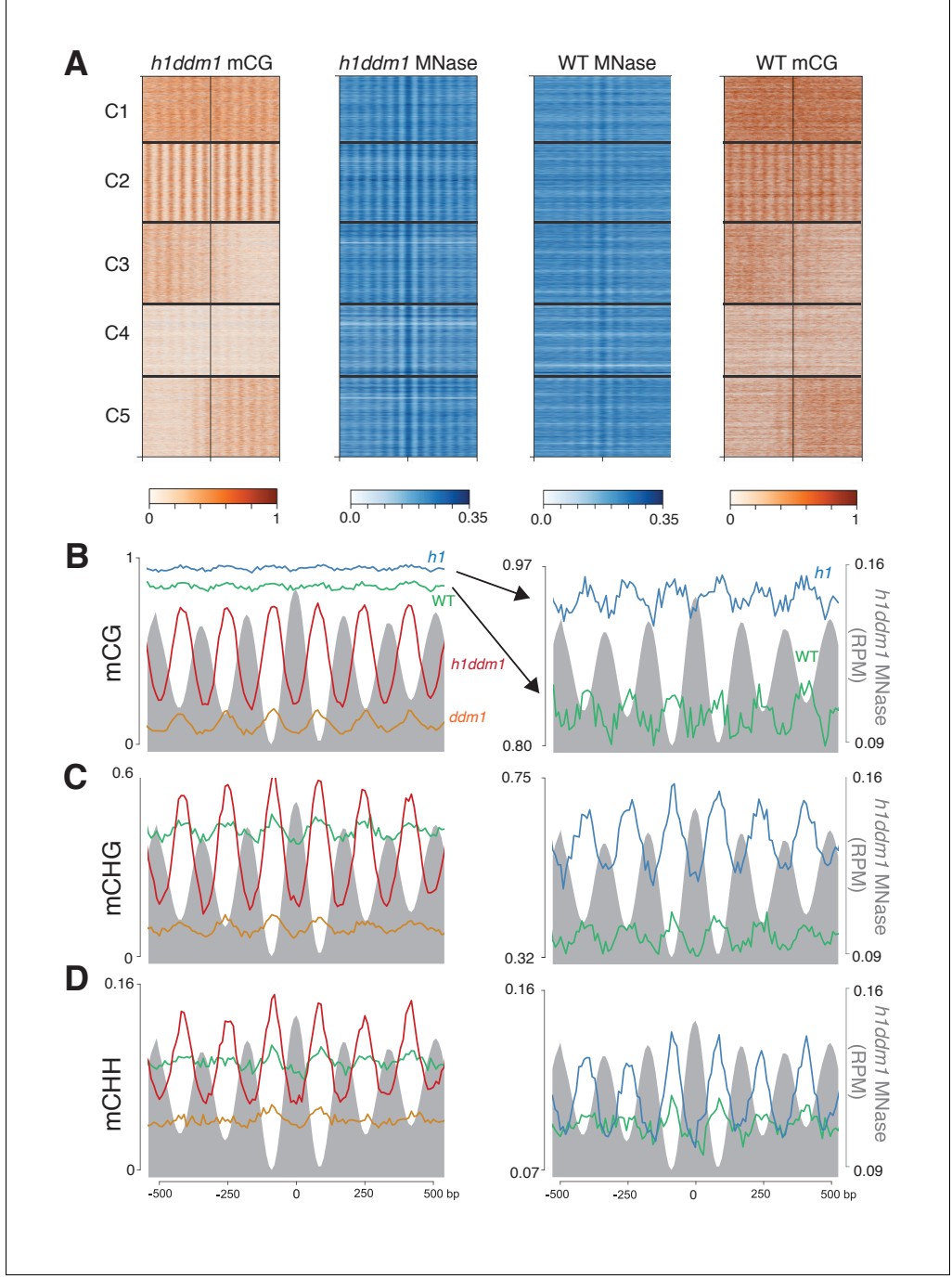

**Figure 5.** Loss of DDM1 and H1 produces linker-specific DNA methylation in *Arabidopsis*. (**A**) *h1ddm1* CG methylation (mCG) centered on heterochromatic well-positioned nucleosomes (n = 22,745) was grouped into five clusters by a self-organizing map algorithm. Nucleosomes (blue) and mCG (orange) are shown for *h1ddm1* and WT; denser shades indicate increased values. (**B–D**) Average methylation plots for all genotypes in the indicated context at cluster C2, with *h1ddm1* nucleosomes in grey fill. WT and *h1* methylation plots are shown zoomed in on right panels, as indicated by arrows. *Figure 5—figure supplement 1* contains non-CG methylation heatmaps. *Figure 5—figure supplement 2* shows the phylogenetic distribution of species with naturally-occurring linker-specific DNA methylation. *Figure 5—figure supplement 3* illustrates the absence of canonical linker histone in most species with linker-specific DNA methylation.

DOI: https://doi.org/10.7554/eLife.30674.014

The following figure supplements are available for figure 5:

*Figure 5 continued on next page*

*Figure 5 continued*

**Figure supplement 1.** *h1ddm1* and WT non-CG methylation at well-positioned arrays of nucleosomes in *h1ddm1* plants.
DOI: https://doi.org/10.7554/eLife.30674.015
**Figure supplement 2.** Linker-specific DNA methylation has evolved multiple times.
DOI: https://doi.org/10.7554/eLife.30674.016
**Figure supplement 3.** Eukaryotic histone H1 protein sequence alignment in species with and without nucleosomal core DNA methylation.
DOI: https://doi.org/10.7554/eLife.30674.017

---

*2005*) and could explain why heterochromatic nucleosomes – and not just linkers – show higher methylation in *h1* compared to WT, and in *h1ddm1* compared to *ddm1* (*Figures 1B* and *5B–D*).

The likely involvement of multiple remodelers in euchromatic DNA methylation does not by itself explain the robust enrichment of methylation within exonic nucleosomes (*Figure 3*). The origins and biological functions of gene body methylation remain quite mysterious (*Zilberman, 2017*), and therefore any explanation of its features is inherently speculative. Given that introns do not exhibit preferential methylation of nucleosomes (*Figure 3C*), the pathways that target methylation to genes are unlikely to favor nucleosomes, especially considering that methylation in genes and TEs is catalyzed by the same methyltranferases (*Law and Jacobsen, 2010*). Preferential methylation of nucleosomes within exons may be related to the function of gene body methylation. DNA methylation is known to affect the properties of nucleosomes, such as positioning and stability (*Davey et al., 1997*; *Choy et al., 2010*; *Jimenez-Useche et al., 2013*; *Huff and Zilberman, 2014*). If gene body methylation influences splicing, other aspects of RNA processing, or the stability of transcript elongation (*Zilberman, 2017*), it may do so in part by altering the properties of exonic nucleosomes. Enrichment of methylation over exonic nucleosomes may therefore be a product of selection. Despite their preferential methylation, genic nucleosomes – at least those experiencing low levels of transcription – are refractory to MET1 methyltransferase activity, as evidenced by their decreased methylation in *ddm1* mutants (*Figure 3D*).

The ability of nucleosomes and H1 to block DNA methylation in the absence of remodeling is dramatically illustrated by a set of heterochromatic loci with arrays of positioned nucleosomes (*Figure 5*, *Figure 5—figure supplement 1*). Here, linker methylation that is close to or even above WT in *h1ddm1* plants is juxtaposed with severe depletion of nucleosomal methylation. Methylation

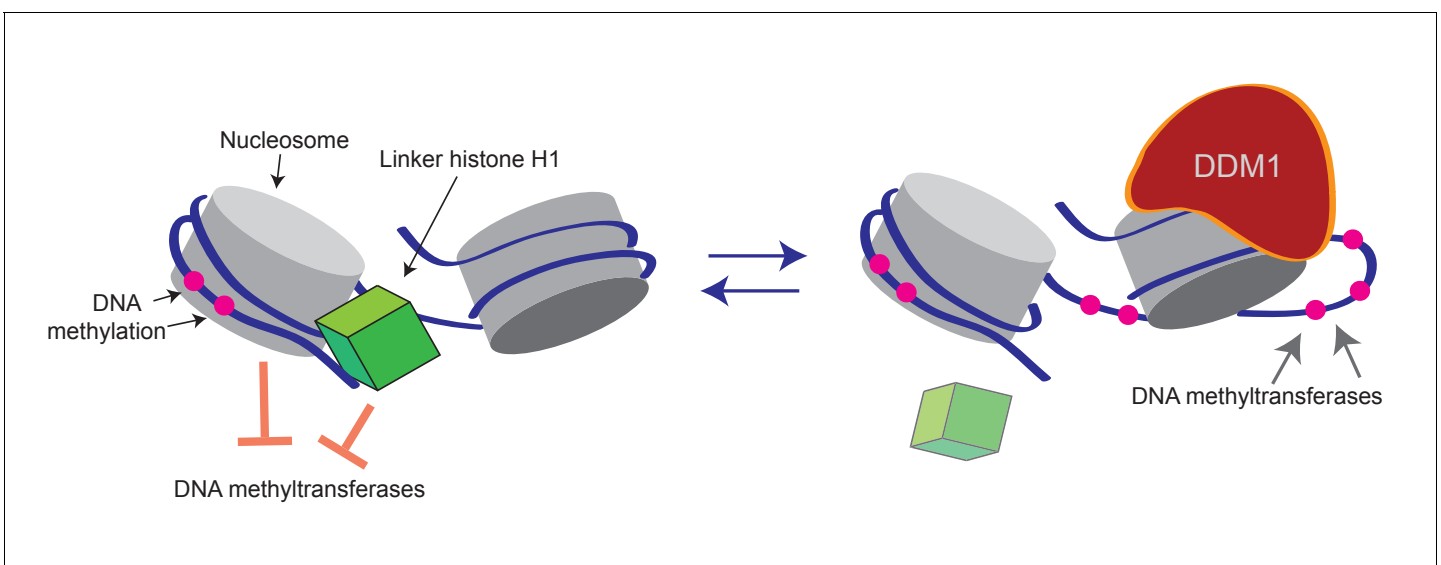

**Figure 6.** Model of proposed role for H1 and DDM1/Lsh in DNA methylation. Cartoon depicts a linker histone-bound dinucleosomal fragment of the genome (left) undergoing remodeling by DDM1 (right) to provide methyltransferases access to nucleosomal core DNA.
DOI: https://doi.org/10.7554/eLife.30674.018

patterns at these loci closely resemble the natural patterns of several diverse species in which methylation is confined to linker DNA throughout the genome (*Huff and Zilberman, 2014*). This resemblance is unlikely to be coincidental. Species with linker-specific methylation are interspersed on the tree of life with species that do not limit methylation to linkers (*Figure 5—figure supplement 2*; *Huff and Zilberman, 2014*). This evolutionary history has been difficult to explain, particularly if one assumes that transitions to or from linker-specific methylation require changing the intrinsic ability of DNA methyltransferases to access nucleosomes.

Our ability to impose an essentially linker-specific methylation pattern on a substantial fraction of the *Arabidopsis* genome by inactivating two chromatin proteins provides a plausible solution to this quandary. Species in which linker-specific methylation has been described have well-positioned nucleosomes (*Huff and Zilberman, 2014*) and all but one of these species (*Micromonas pusilla*) lack the winged-helix motif of canonical H1 (*Figure 5—figure supplement 3*) (*Kasinsky et al., 2001*; *Worden et al., 2009*). A transition to linker-specific methylation in a species with reliably positioned nucleosomes and low or absent linker histones would only require the uncoupling of nucleosome remodeling from DNA methylation maintenance. This could occur through inactivation of the relevant remodeler, or – more likely – by a mutation that severs the spatiotemporal connection between DNA methylation and remodeling. Such a mutation, if sufficiently advantageous, could also be fixed in a species with complex nucleosome positioning and high H1. The linker-specific methylation patterns almost certainly offer a selective advantage to the species that have them, in part by contributing to nucleosome positioning (*Huff and Zilberman, 2014*). Thus, loss of H1 could be advantageous in the presence of a linker-specific methylation system, and reliable nucleosome positioning a consequence of such a system. Considering the many independent instances of complete DNA methylation loss among eukaryotes (*Zemach and Zilberman, 2010*), fixation of advantageous mutations that separate methylation from remodeling could plausibly occur multiple times during eukaryotic evolution. Such a model would easily explain the diverse methylation patterns observed among eukaryotes by providing the mechanism by which apparently radically different patterns can repeatedly evolve.

## Materials and methods

### Biological materials

*ddm1-10* (*Zhang et al., 2016*), *h1.1* and *h1.2* (*Zemach et al., 2013*) *Arabidopsis thaliana* (Columbia ecotype) were crossed to generate *h1ddm1* and siblings for this study. Mutation of DDM1 causes permanent methylation losses at many loci that are not recovered after restoration of DDM1 activity (*Teixeira et al., 2009*). We therefore used the *ddm1-10* allele, which, unlike the *ddm1-2* allele we used in a previous study (*Zemach et al., 2013*), was not homozygous at any point prior to the generation of the compound *h1ddm1* mutant reported here. For MNase-seq, RNA-seq, and bisulfite-seq samples, plants were germinated on agarose plates supplemented with Gamborg's B-5 growth media (GBP07, Caisson Labs). Following germination, they were transferred to soil and grown on a 16hr-light/8hr-dark long-day schedule. For all experiments described here, rosette leaves of 1 month old plants were used.

### Leaf genomic DNA isolation, bisulfite conversion, and sequencing library preparation

Genomic DNA (gDNA) was extracted from 1-month-old *Arabidopsis thaliana* rosette leaves with the DNeasy plant mini kit (Qiagen, cat. no. 69104) per the manufacturer's instructions. Libraries were prepared from roughly 500 ng of purified gDNA that was sheared to approximately 400 bp on a Misonix water bath sonicator, then purified using 1.2X volume of Agencourt Ampure beads (referred to as 'beads' henceforth, cat. no. A63881). Following ligation of methylated Truseq sequencing adapters (Illumina Hayward, CA), bisulfite conversion of DNA was carried out according to manufacturer's protocol (Qiagen Epitect Kit, cat. no. 59104) except without using carrier RNA. DNA was purified twice with 1.2X beads and converted a second time to ensure complete bisulfite conversion of unmethylated cytosine. NEB next indexing primers (cat. no. E7335S) were used for generating multiplexed libraries during PCR amplification of libraries.

## Leaf RNA isolation and library preparation

Leaves were flash frozen in liquid nitrogen, pulverized with mortar and pestle on dry ice, and the resulting material was subjected to vortexing in Trizol (Invitrogen, cat. no. 15596–026). Chloroform was then added at one-fifth the total volume and further vortexing was carried out until the solution appeared homogenous. RNA was subsequently pelleted in ice-cold isopropanol. The resuspended RNA was subjected to rRNA removal with Ribo-zero plant kit (Illumina, MRZPL1224) according to the manufacturer's protocol. 50 ng of ribo-depleted RNA was used for library preparation with the Scriptseq kit (Epicentre, cat. no. SSV21124) following the manufacturer's protocol but with the following modifications: the RNA fragmentation step was extended to 10 min, and the temperature was increased to 90°C.

## MNase digestion of isolated leaf nuclei and sequencing library preparation

Approximately 1 g of pulverized flash frozen rosette leaf tissue from F4 generation plants was resuspended on ice in nuclei isolation buffer (0.25 M sucrose, 15 mM PIPES pH 6.8, 5 mM $MgCl_2$, 60 mM KCl, 15 mM NaCl, 0.9% Triton X-100, 1 mM PMSF, 1X protease inhibitor cocktail) and allowed to thaw for 15 min. The suspension was then strained once through Miracloth (EMD-Millipore, cat. no. 475855) and the nuclear pellet was centrifuged in a swinging-bucket rotor at 2000 g in pre-chilled 5 ml conical tubes. Buffer was carefully aspirated from the loose pellet, which was subsequently resuspended in TM2 (10 mM Tris–HCl, pH 8, 2 mM $MgCl_2$) and washed twice in TM2. The pellet was then resuspended in pre-warmed MNase digestion buffer (50 mM Tris-HCl, pH 8, 5 mM $CaCl_2$) and 25 units of MNase were added per 0.1 g initial starting material and incubated for 10 min at 37°C with periodic agitation; reactions were stopped with EGTA (20 mM final concentration). This was followed by the addition of SDS to a final concentration of 0.625%, increasing the temperature to 75°C, addition of RNase A for 10 min, and the addition of proteinase-K for an additional 15 min. DNA was then extracted with phenol/chloroform, pelleted in 70% EtOH, then resuspended in TE pH 8.0.

Verification of MNase digestion was carried out by fractionating one-tenth of the total reaction on a standard TAE gel and checking for a majority mononucleosomal fraction. Following removal of short DNA with beads, libraries were synthesized using total DNA (without PCR amplification) using the Nugen Ovation kit (Cat. no. 0319) following the manufacturer's protocol.

## Sequencing

All sequencing was carried out as single-end 100 bp reads, except MNase libraries, which were sequenced in paired-end 100 bp reads on Illumina HiSeq 2500 or −4000 at the QB3 Vincent Coates Genomic Sequencing Lab at UC Berkeley.

## Short read alignment

Bisulfite sequencing reads were mapped with Bowtie (*Langmead et al., 2009*) using the bs-sequel pipeline (available at http://dzlab.pmb.berkeley.edu/tools), with Bowtie settings allowing for up to two mismatches in the seed and reporting up to 10 matches for a given read. 100 bp paired-end MNase-seq reads were mapped using Bowtie2 (*Langmead and Salzberg, 2012*) allowing for up to one mismatch in the seed region and only the best match reported. All alignments were made to the TAIR10 *Arabidopsis thaliana* genome assembly.

## RNA-seq analysis

The Kallisto (*Bray et al., 2016*) quant command was invoked with the following settings: –single –fr-stranded -b 100 l 320 s 30. Kallisto output was fed into the R environment for processing with the Sleuth software package (*Pimentel et al., 2017*), which was used to perform expression quantification and derive expression deciles.

## Nucleosome identification and classification

For initial identification of nucleosomes, the Python program iNPS (*Chen et al., 2014*) was used on our paired-end MNase-seq reads (120 to 180 bp, not inclusive), which were filtered using SAMtools (*Li et al., 2009*). iNPS produces a number of nucleosome types depending on the shape of read distribution: only 'main peaks' from this output were taken from our biological replicates and used in

identification of positioned nucleosomes. The other output from this program, the 'like_Wig' files, were used in visualization of MNase-seq data in the genome browser as well as for calculating average nucleosome enrichment across loci.

We took iNPS nucleosome peaks and removed all that were wider than 140 bp. Remaining nucleosome overlap was calculated using the built-in '-wo' option of bedtools (*Quinlan and Hall, 2010*) 'intersect' function. The percent overlap of peaks from replicate one with peaks from replicate two was then used to define regions of the genome with reliably positioned nucleosomes. Nucleosome peaks that overlapped reciprocally more than 75% were classified as well-positioned (left-most panel in *Figure 1A*, for example). Less-well-positioned nucleosomes were further defined in 25% increments of overlap, such that the next category overlapped more than 50% but less than 76%, and so on, generating four groups, plus a group that was comprised of other nucleosomes. To isolate differentially-methylated regions (DMR) outside of positioned nucleosomes (see below for definition of DMRs), we used DMRs not overlapping any of the four groups defined above.

The presumptive centers of these nucleosomal loci were calculated as the arithmetic means of the combined 5'- and 3'-most ends of the overlapping nucleosomes, allowing for multiple overlaps for a given nucleosome. Nucleosome groups per genotype are provided in GEO (GSE96994). Shared nucleosomes (*Figure 1—figure supplement 2*) are defined as those nucleosomes whose calculated dyad is within 20 bp (inclusive) of the other genotype. For *h1ddm1*, three biological replicates were generated for visualization but only two were used as input to determine nucleosome position because addition of a third replicate did not improve the positioning analysis. The third replicate is available in the GEO accession associated with this work.

## Description of *Arabidopsis* genome features

'Heterochromatic sequences' in *Figure 1* refers to merged TAIR10 TEs exhibiting >5% CG methylation that are in the upper two quintiles for H3K9me2 enrichment (as calculated from *Stroud et al., 2014*) and are longer than 30 bp (complete merged TE bed file provided in GEO accession GSE96994). TEs referred to related to *Figure 3* are all merged TEs, regardless of H3K9me2 level. The TSS (*Figure 3A*) was defined as ±1 kb from the annotated TAIR10 TSS; while gene body was considered the portion downstream of 1 kb. Genes were defined to exclude annotated TEs or sequences methylated like TEs, as in (*Zemach et al., 2013*).

## Definition of DRM1/2 and CMT2 dependent differentially-methylated regions (DMRs)

50 bp windows with at least 10% WT CHH methylation (mCHH) and with more than 30% loss of mCHH in *drm1drm2* or *cmt2* were used to define DRM1/2 and CMT2 dependent DMRs, respectively. These windows also exhibited statistically significant decrease in mCHH from WT ($p < 0.01$, Fisher's exact test). We further filtered out DMRs that exhibited significant mCHH differences relative to WT in both genotypes, ensuring the DMRs are exclusive to the given genotype. After merging adjacent 50 bp DMR windows, there were 54,106 CMT2 DMRs and 8640 DRM1/2 DMRs. These coordinates are available in GEO accession GSE96994. All DMRs overlap at least one annotated TE.

Methylation plots in *Figure 2* are centered either on 'group1' genotype-specific well positioned nucleosomes that overlap the indicated DMR by at least 1 bp or, for the 'DMR' plots in the same figure, we used the arithmetic center of DMRs that do not overlap with any of the nucleosomes identified in groups 1 through 4, also in a genotype-specific manner (e.g. *h1* DMR plots are anchored to those DMRs which do not overlap any of the *h1* group 1–4 nucleosomes).

## Histone H1 protein alignment

We aligned sequences with T-coffee (*Notredame et al., 2000*) in normal mode and plotted the resulting alignment file with Boxshade (http://www.ch.embnet.org/). The following sequences were used to perform multiple sequence alignment shown in *Figure 5—figure supplement 3*: *Aureococcus anophagefferens*, protein ID 72830, scaffold_256:8187–9341; *Ostreococcus lucimarinus*, protein ID 18993, Chr_20:191746–192084 (+); *Emiliana huxleyi* protein ID 448806/460675, scaffold_1716:1449–2181 (+)/scaffold_11:31908–32637 (-); *Micromonas pusilla*, protein ID 1714, scaffold_11:410445–411786, *Arabidopsis thaliana* H1.1 (AT1G06760), *Neurospora crassa* hH1 (ORF name NCU06863), and *Mus musculus* H1a (MGI:1931523).

## Use of previously published data

CTCF sites (*Baubec et al., 2015*) were used as anchors for plotting averaged DNA methylation and MNase-seq data. Lsh mutant and WT MEF DNA methylation data (*Yu et al., 2014*) and WT MNase-seq data (*Teif et al., 2012*) were obtained through GEO (GSE56151 and GSE40896, respectively). Bin widths for methylation and MNase analysis were both set at 10 bp.

## Calculation of FFT for nucleosomal DNA

For *Arabidopsis*, 147 bp MNase-seq fragments were isolated from one of the biological replicates for each genotype. The resulting reads were used as anchors to calculate mean DNA methylation using bisulfite data (*Chodavarapu et al., 2010*); this methylation vector was used as input to the TSA R program (function 'periodogram') (*Chan and Ripley, 2012*), which generated the power spectrum over a range of frequencies. 'spec' output was plotted against the inverse of frequency (in this case, bp), truncated to 30 bp and plotted. For mouse, 147 bp fragments from *Teif et al. (2012)* that overlapped CTCF sites (±1 kb) (*Baubec et al., 2015*) were used as anchors for methylation averaging, and the strength of periodicity was calculated as above.

## Data visualization

For MNase-seq plotting, in order to compare reads mapped across different genotypes and multiple replicates, we normalized by multiplying the smoothed output of iNPS by the reads per million per nucleotide (RPM). For instance, a 10 bp bin score of MNase-seq in sample A would be multiplied by $1 \times 10^6$/(number of mapped reads in sample A). Perl scripts (http://dzlab.pmb.berkeley.edu/tools) were used to generate enrichment score matrices of mapped data around genomic features of interest. These matrices were imported to R (*R Core Team , 2017*; *Davey et al., 1997*) for further processing and visualization using base plotting functions and the ggplot2 library (*Wickham, 2009*). Heatmaps were generated with Seqplots (*Stempor et al., 2016*) using the self-organizing map (SOM) clustering algorithm. Genome tracks are manual screenshots of our data displayed in IGV (*Robinson et al., 2011*).

# Acknowledgements

We thank X Feng for helpful comments on the manuscript and T Baubec and D Schubeler for the list of CTCF binding sites, as well as P-H Hsieh for help in defining DMRs. We also thank S McDevitt for help with sequencing and C Wistrom for greenhouse assistance. This work was supported by a fellowship from the Helen Hay Whitney Foundation and NIH grant T32HG000047 to DBL, and European Research Council grant MaintainMeth (725746) and a Faculty Scholar grant from HHMI and the Simons Foundation (55108592) to DZ. Data generated from this study is available under GEO accession GSE96994. The authors declare no competing financial interests.

# Additional information

## Competing interests

Daniel Zilberman: Reviewing editor, *eLife*. The other author declares that no competing interests exist.

## Funding

| Funder | Grant reference number | Author |
|---|---|---|
| Howard Hughes Medical Institute | 55108592 | Daniel Zilberman |
| Horizon 2020 Framework Programme | 725746 | Daniel Zilberman |
| National Institutes of Health | T32HG000047 | David B Lyons |
| Helen Hay Whitney Foundation | | David B Lyons |

The funders had no role in study design, data collection and interpretation, or the decision to submit the work for publication.

## Author contributions

David B Lyons, Conceptualization, Data curation, Formal analysis, Funding acquisition, Investigation, Visualization, Writing—original draft, Writing—review and editing; Daniel Zilberman, Conceptualization, Formal analysis, Supervision, Funding acquisition, Writing—original draft, Writing—review and editing

## Author ORCIDs

David B Lyons http://orcid.org/0000-0002-5721-4080
Daniel Zilberman http://orcid.org/0000-0002-0123-8649

## Decision letter and Author response

Decision letter https://doi.org/10.7554/eLife.30674.027
Author response https://doi.org/10.7554/eLife.30674.028

# Additional files

## Supplementary files

• Transparent reporting form
DOI: https://doi.org/10.7554/eLife.30674.019

## Major datasets

The following dataset was generated:

| Author(s) | Year | Dataset title | Dataset URL | Database, license, and accessibility information |
| --- | --- | --- | --- | --- |
| Lyons DB, Zilberman D | 2017 | DDM1 and Lsh remodelers allow methylation of DNA wrapped in nucleosomes | https://www.ncbi.nlm.nih.gov/geo/query/acc.cgi?acc=GSE96994 | Publicly available at the NCBI Gene Expression Omnibus (accession no: GSE96994) |

The following previously published datasets were used:

| Author(s) | Year | Dataset title | Dataset URL | Database, license, and accessibility information |
| --- | --- | --- | --- | --- |
| Yu W, McIntosh C, Lister R, Zhu I, Han Y, Ren J, Landsman D, Lee E, Briones V, Terashima M, Leighty R, Ecker JR, Muegge K | 2014 | CG hypomethylation in Lsh-/- mouse embryonic fibroblasts is associated with de novo H3K4me1 formation and altered cellular plasticity | https://www.ncbi.nlm.nih.gov/geo/query/acc.cgi?acc=GSE56151 | Publicly available at the NCBI Gene Expression Omnibus (accession no: GSE56151) |
| Teif VB, Vainshtein Y, Caudron-Herger M, Mallm J, Marth C, Höfer T, Rippe K | 2012 | Genome-wide nucleosome positioning during embryonic stem cell development | https://www.ncbi.nlm.nih.gov/geo/query/acc.cgi?acc=GSE40896 | Publicly available at the NCBI Gene Expression Omnibus (accession no: GSE40896) |

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
