## [Decision Letter]

Thank you for submitting this interesting study "DDM1 and Lsh remodelers allow methylation of DNA wrapped in nucleosomes" for consideration by *eLife*. Your article has been favorably evaluated by Detlef Weigel (Senior Editor) and three reviewers, one of whom, Richard Amasino (Reviewer #1), is a member of our Board of Reviewing Editors. The following individual involved in review of your submission has agreed to reveal their identity: Vincent Colot (Reviewer #2).

The reviewers have discussed the reviews with one another and the Reviewing Editor has drafted this decision below to help you prepare a revised submission.

In the consolidated review below, some parts are directly from the first round of review and a ** was added to those parts when a comment to address was within a paragraph.

In this manuscript, Lyons and Zilberman examine the implication of the chromatin remodeler DDM1 and canonical histone H1 in the relationship between nucleosomes and DNA methylation in Arabidopsis. In previous work, the Zilberman lab showed that removal of canonical histone H1 leads to an increase in DNA methylation over heterochromatic TEs and a partial rescue of the well-documented loss of DNA methylation seen in ddm1. These observations lead to the conclusion that DDM1 provides DNA methyltransferases (MTases) with access to H1-containing heterochromatin (Zemach et al., Cell 2013).

Numerous in vitro and in vivo studies using mammalian MTases and de novo DNA methylation assays have indeed established that the nucleosome core is an obstacle to DNA methylation. The authors should cite Felle et al., NAR 2011 as well as Baubec et al., Nature 2015 in their introductory paragraph (Baubec et al. is only cited later, in a different context). In addition, the authors should cite Takeshima et al., J. Mol. Biol. 2008; in this paper it was shown that histone H1 blocks DNA methylation of linker DNA in vitro.

Lyons and Zilberman used MNase-seq and BS-seq to investigate directly DNA methylation in relation to nucleosome position in h1, ddm1 and h1ddm1 mutant plants as well as in WT controls. Key to their analysis is the definition of four groups of nucleosomes, from well-positioned to poorly-positioned. This distinction enabled the authors to report several new findings, which clarify the role of DDM1 in the maintenance of DNA methylation over H1-enriched nucleosomes.

1) First, they show that heterochromatic nucleosomes, which are enriched in histone H1, have invariably high average CG methylation across the core and linker in WT controls. This is strong evidence that, in vivo, core and linker DNA are equally accessible to MET1, which is responsible for the maintenance of CG methylation. In marked contrast, average CHG and CHH methylation, which is much lower than average CG methylation, is preferentially located over linker DNA for these same nucleosomes. Thus, the MTases responsible for the perpetuation of this non-CG methylation across cell divisions preferentially access linker DNA in vivo.

2/3) The second and key set of findings in this manuscript is that at presumably the same (**this is not clear from the text) well-positioned heterochromatic nucleosomes, the increase and decrease in average CG, CHG and CHH methylation observed in h1 and ddm1 mutant plants, is most and least pronounced over linker DNA, respectively. As could then be predicted, the partial rescue of methylation in the double mutant is also more pronounced over linker DNA. These findings are entirely consistent with the inhibitory effect in vitro of histone H1 on linker DNA methylation (Takeshima et al., J. Mol. Biol. 2008) and as noted above Takeshima et al. should be cited in this context. In addition, they suggest that DDM1 facilitates – to varying extents – the access of the different MTases responsible for CG, CHG and CHH methylation: not only to the core, but also to the linker, of H1-enriched, heterochromatic nucleosomes. **However, in the absence of genomic binding data for these MTases in WT and mutant backgrounds, indirect effects cannot be ruled out. This notwithstanding, the authors also used published nucleosome positioning and methylome data to show that loss of Lsh, the mammalian homolog of DDM1, causes a much greater depletion of DNA methylation over nucleosome cores than over linkers in mouse embryonic fibroblasts.

4) Coming back to Arabidopsis, the authors then show that in contrast to TEs (which comprise most heterochromatic nucleosomes), genes have higher average CG methylation over nucleosome cores than over linker DNA in WT plants. This pattern is evident for exons but not for introns (**an intriguing observation, which would merit being discussed by the authors) and it remains largely unaltered in the three mutant backgrounds, except in the case of lowly expressed genes, perhaps because these genes undergo less transcription-coupled nucleosome displacement (it is not clear though if these genes are not part of TEs). The 10 bp methylation periodicity seen over nucleosome cores in all sequence contexts is also analyzed and results indicate that in the case of heterochromatic nucleosomes, it is dependent on DDM1. Based on this last result, the authors propose a model whereby DDM1 acts by releasing a loop of DNA from the surface of the histone octamer.

In a last part, the authors show that a subset of heterochromatic nucleosomes that are well-positioned in the h1ddm1 double mutant exhibit the same DNA methylation pattern as seen in marine algae, namely a perfect alternation of low and high methylation over the nucleosome core and linker, respectively.

In sum, this manuscript reports many new findings that are of broad interest. It adds nicely to the two other comprehensive studies by the Zilberman lab (Zemach et al., Cell 2013; Huff and Zilberman 2014) and to that of Baubec et al. (Nature 2015) indicating that nucleosomes inhibit DNA methylation in vivo. In addition, it provides novel and compelling evidence that the linker histone H1 inhibits DNA methylation in vivo mainly by preventing access to linker DNA, consistent with in vitro data (Takeshima et al., J. Mol. Biol. 2008). Thus, as the authors suggest, the distinct patterns of DNA methylation observed among eukaryotes are likely determined in large part by whether or not a given species has well positioned nucleosomes and possesses the linker histone H1 and/or chromatin remodelers that can provide DNA MTases access to nucleosomes.

5/6) Overall, the data as well as the analyses presented in the manuscript are solid and comprehensive. **However, most analyses are described too succinctly and it is often difficult to understand what exactly was done (see below for a list of specific points that the authors need to address in addition to those raised in the general comments). Furthermore, the manuscript is difficult to read in places and I would recommend that the authors **subdivide it into three clearly identified sections (Introduction, Results, Discussion) as well as several sub-sections in Results. This division will also help separate what is a fair interpretation of the findings reported in the manuscript, from the more speculative conclusions.

7) One of the authors' main claims is that mutation of both DDM and Lsh1 bias methylation towards linker DNA. To support, and fully evaluate, this claim the following points should be addressed:

a) The conclusion from Figure 1 is that nucleosomes are refractory to DNA methylation without chromatin remodelers. However, it is unclear how the correlations between DNA methylation and nucleosomes were conducted. For panels A and C, either the methylation in all the mutants is shown relative to the nucleosome position in WT plants, in which case potential differences in nucleosome position in the mutants should be accounted for, or the methylation is shown relative the nucleosome positioning in each mutant, in which case the sequence differences in the nucleosome cores vs linker regions should be accounted for. Likewise, for the mammalian data, where the methylation data in the WT and lsh mutants is relative to the nucleosome positions in WT samples, changes in nucleosome positions in the lsh mutant should be accounted for in the analysis. Without such information it remains unclear whether the differences in methylation observed in the mutants are associated with changes in nucleosome positions, DNA sequence biases, and/or changes in the ability of DNA methyltransferases to methylate nucleosomal or linker DNA.

b) For the lsh mutant, a meta-analysis is presented showing decreases in DNA methylation over well positioned nucleosomes and near WT levels of methylation in the linker regions; thus, methylation is skewed towards linkers. For ddm1/h1ddm1 mutants a similar trend is observed. However, more in-depth analyses should be presented to strengthen the assertion that LSH and DDM1 are acting similarly. For example, is Lsh activity also mainly associated with heterochromatin? Is the methylation periodicity in mammalian cells disrupted in lsh mutants?

8) Another major point is that DDM1 enables the methylation of nucleosomal DNA by remodeling the chromatin such that DNA methylation actually occurs on a DNA loop and not directly on nucleosome bound DNA.

Here the authors suppose (without any data) that this loop would still retain a bias in DNA methylation resulting in a periodic pattern matching previous in vivo findings. To test their model, the periodicity of DNA methylation was then assessed in a ddm1 mutant background. However, as this mutant causes global decreases in DNA methylation, disruption of the periodic behavior is hard to assess. Furthermore, the decrease in periodicity could either be because the DNA loops aren't created so methylation is blocked or because DDM1 serves a different function that allows MET1 to directly methylate nucleosomal DNA. Thus, these in vivo experiments remain inconclusive. To support their conclusions, in vitro remodeling assays showing that DDM1 activity creates accessible DNA loops that can be methylated by MET1 would be highly informative, although such assays are clearly not trivial. In the absence of such assays, the authors should note the alternative interpretations noted above.

9) A final point of the paper is that the decoupling of chromatin remodeling and DNA methylation could explain some of the differences in global methylation patterns observed between species.

Here the authors find that a subset of well positioned, heterochromatin-localized nucleosomes (cluster C2) have strong linker methylation and low nucleosome methylation in the h1ddm1 background. Based on the similarity of this DNA methylation pattern with those observed in algae, it is proposed that decoupling methylation and remodeling is a mechanism employed to generate such patterns in organisms with well positioned nucleosomes. However, this model is not further vetted. In this case, a phylogenetic analysis correlating the presence or absence of DDM1 or H1 orthologs with linker-specific methylation patterns might provide additional support for the authors claims.

Although in general we appreciate brevity, some elaboration here would be useful to the reader. Do you think this phenomenon is a result of genetic drift in which in some organisms the loss of one component is not selected against or might it provide some selective advantage? In this section is the sentence: "These species are interspersed on the tree of life with species that do not limit methylation to linkers (Figure 4—figure supplement 1)." Figure 4—figure supplement 1 (does this mean Figure 4—figure supplement 1?) does not seem to have much relation to the sentence that cites them – i.e., nothing to do with the phylogenetically "interspersed" nature of the phenomenon because both figures are about Arabidopsis.

Specific comments:

1) Main text, first paragraph: Replace Jones and Baylin, 2007 with Baylin and Jones, CSH Persp. Biol. 2016, which is more appropriate and recent.

2) Main text, first paragraph: To be consistent with the first part of the sentence, I would be more general here and provide a review instead of Ong-Abdullah et al., 2015.

3) Main text, fourth paragraph: It is not clear if the four groups of nucleosomes were determined for each of the four genotypes or solely for the WT and then maintained for the analysis of the three mutant genotypes? I suspect that it is the latter, but the description in Materials and methods suggests otherwise. How can meaningful comparisons be made if the four groups of nucleosomes differ for each genotype?

4) The authors could easily complement their analysis of heterochromatic nucleosomes by considering separately those that have CMT2-dependent CHH methylation and those that are targeted by DRM1/2 instead. This additional analysis could also prove very informative.

5) Main text, fourth paragraph: Make also reference to the Materials and methods section ("Description of Arabidopsis genome features").

6) Main text, seventh paragraph: “Figure 2—figure supplement 1” should be Figure 1—figure supplement 2.

7) Main text, ninth paragraph: Are all genes non TE-genes? And what about TEs? Please provide about how the TE class was defined and how it differs from the heterochromatic nucleosomes analyzed in the first part.

8) Main text, tenth paragraph: The fact that gene methylation is enriched over nucleosomes and that DDM1 is not required for this type of methylation except in the case of lowly expressed genes suggests that other chromatin remodelers are involved. However, at this stage this is only a hypothesis and the authors should therefore tone down the last sentence of this paragraph.

9) Materials and methods:

- Why was the mutant allele ddm1-10 chosen over the much more commonly studied allele ddm1-2, used for instance in Zemach et al., 2013.

- Which generation was used for ddm1 and h1ddm1? This is important to know as the demethylation of heterochromatic sequences increases with the number of generations.

- Were DNA, RNA and nucleosomes extracted from the same starting biological material?

10) Figure 1 and Figure 1—figure supplement 2: I would recommend that the supplement be moved to Figure 1, as it is clear that CHH methylation is also affected in the three mutant backgrounds. I would also recommend for each of the three sequence contexts merging the four graphs, so as to make more apparent the gains and losses of methylation. This is particularly important for CHH methylation, which is low to start with compared to CHG and especially CG methylation and which show modest fluctuations in absolute values. Indicating levels relative to WT could improve visualization further.

11) The legend of Figure 4 is confusing as a result that figure is difficult to understand. The authors should also show mCHG and mCHH data or else justify why they chose not to show them.

In Figure 4 it is not clear what data is being shown in the right panels.

12) It is not clear from the methods if the same stage leaf tissue was used for the plant DNA methylation and nucleosome profiling experiments. As these features can vary during development, this point should be clarified.

[Editors' note: further revisions were requested prior to acceptance, as described below.]

Thank you for resubmitting your work entitled "DDM1 and Lsh remodelers allow methylation of DNA wrapped in nucleosomes" for further consideration at eLife. Your revised article has been favorably evaluated by Detlef Weigel (Senior Editor), a Reviewing Editor, and two reviewers.

There are a few minor points to address that the reviewers and I think would make your paper even more clear to the eLife readership.

1)We suggest a more direct formulation. "DNA methylation at DRM1/2-dependent loci is actually enriched around well-positioned nucleosomes in WT (Figure 2). This enrichment is likely caused by the small size of DRM1/2-dependent TEs and hence the high proportion of non-methylated adjacent sequences over these nucleosomes. Consistent with this interpretation, an analysis anchored to the centers of differentially methylated regions (DMRs; see Methods)[…]"

2) Discussion, fourth paragraph, last sentence: This sentence is ambiguous. Gene body methylation is affected by MET1, with no contribution from the other methyltransferases. The ddm1 mutation affects only a small subset of body methylated genes.

3) Re the statement in the Abstract "We find that DDM1 enables methylation of DNA bound to the nucleosome, thus demonstrating that nucleosome-free DNA is the preferred substrate of eukaryotic methyltransferases in vivo." The first half of the statement is well supported, but the data do not demonstrate that nucleosome-free DNA is the preferred substrate. Rather, the data support this hypothesis. It is up to you how to re-phrase, but an example of re-phrasing would be "We find that DDM1 enables methylation of DNA bound to the nucleosome, consistent with a model in which nucleosome-free DNA is the preferred substrate of eukaryotic methyltransferases in vivo."

4) In several cases, you draw specifically on the effects of h1 on CG methylation (see quoted text below). However, the effects of h1 on CG methylation are minimal. Indeed, when plotted relative to WT levels in Figure 1, the line is flat. However, h1 clearly has effect on nucleosome core methylation levels in the CHG and CHH contexts, and it also enhances the ddm1 mutant. In light of these observations the text should be modified to more accurately reflect the data presented.

"CG methylation is relatively evenly distributed with respect to nucleosomes in WT plants, and shows a weak nucleosomal depletion in h1 mutants (Figure 1)."

"Without H1, linker DNA becomes even more accessible, producing the observed slight core-to-linker methylation differential in h1 mutants (Figure 1).”

5) The text now makes clear that the methylation data shown in Figure 1 is relative to the specific nucleosome positions in each genotype. The authors also account for possible affects due to differences in the sequences in the nucleosomes and the linkers through their analysis in Figure 1—figure supplement 2. However, to fully evaluate this data, it would be informative to know how many well positioned nucleosomes there are in total and how many are shared between WT and h1ddm1 or between h1 and h1ddm1. In the response to reviewers it is stated that "only a limited subset of WT well-positioned nucleosomes are also well positioned in the mutants." If this overlap is very small, then it remains unclear if this subset of regions can be used to extrapolate information regarding global effects.

6) The implications of the following statement are unclear.

"However, this appearance is almost certainly caused by the small size of DRM1/2-dependent TEs, because an analysis anchored to the centers of differentially methylated regions (DMRs; see Methods) between drm1drm2 mutants and wild-type that occur in regions with very poorly positioned nucleosomes (those outside the four groups defined in Figure 1; see Methods) shows an even sharper peak (plots in the right column of Figure 2)."

7) To avoid confusion the word "can" should be added to this sentence of the Introduction: "Methylation also regulates endogenous genes: methylation close to the transcriptional start site *can* cause gene silencing[…]"

---

## [Author Response]

In the consolidated review below, some parts are directly from the first round of review and a ** was added to those parts when a comment to address was within a paragraph.In this manuscript, Lyons and Zilberman examine the implication of the chromatin remodeler DDM1 and canonical histone H1 in the relationship between nucleosomes and DNA methylation in Arabidopsis. In previous work, the Zilberman lab showed that removal of canonical histone H1 leads to an increase in DNA methylation over heterochromatic TEs and a partial rescue of the well-documented loss of DNA methylation seen in ddm1. These observations lead to the conclusion that DDM1 provides DNA methyltransferases (MTases) with access to H1-containing heterochromatin (Zemach et al., Cell 2013).Numerous in vitro and in vivo studies using mammalian MTases and de novo DNA methylation assays have indeed established that the nucleosome core is an obstacle to DNA methylation. The authors should cite Felle et al., NAR 2011 as well as Baubec et al., Nature 2015 in their introductory paragraph (Baubec et al. is only cited later, in a different context). In addition, the authors should cite Takeshima et al., J. Mol. Biol. 2008; in this paper it was shown that histone H1 blocks DNA methylation of linker DNA in vitro.

We have added the following sentence to the Introduction section:

“However, mammalian methyltransferase activity is inhibited by nucleosomes and H1 in vitro (Robertson et al., 2004; Takeshima et al., 2006; Takeshima et al. 2008; Felle et al., 2011; Schrader et al., 2015), and regions of the human and mouse genomes with reliably positioned nucleosome arrays organized by CTCF binding exhibit preferential methylation of linker DNA (Kelly et al., 2012; Baubec et al., 2015).”

We are reluctant to cite Felle et al. in support of linker-biased methylation in vivo because the in vivo experiments in this paper are deeply flawed. The authors find that larger MNase-digested fragments, which represent multiple nucleosomes and are enriched for linker DNA, are more methylated than shorter mononucleosome fragments. They infer that linkers are more methylated than nucleosomes, but this assumes all genomic sequences are equally accessible to MNase. Instead, the shorter fragments come from more accessible and less methylated euchromatin, explaining the enrichment of methylation on longer fragments.

Lyons and Zilberman used MNase-seq and BS-seq to investigate directly DNA methylation in relation to nucleosome position in h1, ddm1 and h1ddm1 mutant plants as well as in WT controls. Key to their analysis is the definition of four groups of nucleosomes, from well-positioned to poorly-positioned. This distinction enabled the authors to report several new findings, which clarify the role of DDM1 in the maintenance of DNA methylation over H1-enriched nucleosomes.1) First, they show that heterochromatic nucleosomes, which are enriched in histone H1, have invariably high average CG methylation across the core and linker in WT controls. This is strong evidence that, in vivo, core and linker DNA are equally accessible to MET1, which is responsible for the maintenance of CG methylation. In marked contrast, average CHG and CHH methylation, which is much lower than average CG methylation, is preferentially located over linker DNA for these same nucleosomes. Thus, the MTases responsible for the perpetuation of this non-CG methylation across cell divisions preferentially access linker DNA in vivo.2/3) The second and key set of findings in this manuscript is that at presumably the same (**this is not clear from the text) well-positioned heterochromatic nucleosomes, the increase and decrease in average CG, CHG and CHH methylation observed in h1 and ddm1 mutant plants, is most and least pronounced over linker DNA, respectively. As could then be predicted, the partial rescue of methylation in the double mutant is also more pronounced over linker DNA.

We have clarified the differences between nucleosome sets and added a detailed explanation regarding why we used nucleosomes defined separately for each genotype in Figure 1. We have also added supplemental data to illustrate that our conclusions do not change when analyzing sets of well-positioned nucleosomes shared by different genotypes (Figure 1—figure supplement 2; see point 7a below).

These findings are entirely consistent with the inhibitory effect in vitro of histone H1 on linker DNA methylation (Takeshima et al., J. Mol. Biol. 2008) and as noted above Takeshima et al. should be cited in this context.

We have added a citation to Takeshima et al. (2008).

In addition, they suggest that DDM1 facilitates – to varying extents – the access of the different MTases responsible for CG, CHG and CHH methylation: not only to the core, but also to the linker, of H1-enriched, heterochromatic nucleosomes. **However, in the absence of genomic binding data for these MTases in WT and mutant backgrounds, indirect effects cannot be ruled out.

We have added the following comment to the beginning of our Discussion section:

“It remains formally possible that DDM1/Lsh remodelers permit methyltransferases to work directly on the nucleosome surface or promote catalysis without affecting methyltransferase access.”

This notwithstanding, the authors also used published nucleosome positioning and methylome data to show that loss of Lsh, the mammalian homolog of DDM1, causes a much greater depletion of DNA methylation over nucleosome cores than over linkers in mouse embryonic fibroblasts.4) Coming back to Arabidopsis, the authors then show that in contrast to TEs (which comprise most heterochromatic nucleosomes), genes have higher average CG methylation over nucleosome cores than over linker DNA in WT plants. This pattern is evident for exons but not for introns (**an intriguing observation, which would merit being discussed by the authors) and it remains largely unaltered in the three mutant backgrounds, except in the case of lowly expressed genes, perhaps because these genes undergo less transcription-coupled nucleosome displacement (it is not clear though if these genes are not part of TEs).

We have added a paragraph to the Discussion regarding gene body methylation and exon vs. intron DNA methylation.

The 10 bp methylation periodicity seen over nucleosome cores in all sequence contexts is also analyzed and results indicate that in the case of heterochromatic nucleosomes, it is dependent on DDM1. Based on this last result, the authors propose a model whereby DDM1 acts by releasing a loop of DNA from the surface of the histone octamer.In a last part, the authors show that a subset of heterochromatic nucleosomes that are well-positioned in the h1ddm1 double mutant exhibit the same DNA methylation pattern as seen in marine algae, namely a perfect alternation of low and high methylation over the nucleosome core and linker, respectively.In sum, this manuscript reports many new findings that are of broad interest. It adds nicely to the two other comprehensive studies by the Zilberman lab (Zemach et al., Cell 2013; Huff & Zilberman 2014) and to that of Baubec et al. (Nature 2015) indicating that nucleosomes inhibit DNA methylation in vivo. In addition, it provides novel and compelling evidence that the linker histone H1 inhibits DNA methylation in vivo mainly by preventing access to linker DNA, consistent with in vitro data (Takeshima et al., J. Mol. Biol. 2008). Thus, as the authors suggest, the distinct patterns of DNA methylation observed among eukaryotes are likely determined in large part by whether or not a given species has well positioned nucleosomes and possesses the linker histone H1 and/or chromatin remodelers that can provide DNA MTases access to nucleosomes.5/6) Overall, the data as well as the analyses presented in the manuscript are solid and comprehensive. **However, most analyses are described too succinctly and it is often difficult to understand what exactly was done (see below for a list of specific points that the authors need to address in addition to those raised in the general comments). Furthermore, the manuscript is difficult to read in places and I would recommend that the authors **subdivide it into three clearly identified sections (Introduction, Results, Discussion) as well as several sub-sections in Results. This division will also help separate what is a fair interpretation of the findings reported in the manuscript, from the more speculative conclusions.

We have reorganized the paper into separate Introduction, Results, and Discussion sections. We also added Results subsections and moved more speculative elements of our interpretation of data to the Discussion. We have expanded the explanations of our methods as well as the results, and much of the interpretation of the results has been expanded in the Discussion.

7) One of the authors' main claims is that mutation of both DDM and Lsh1 bias methylation towards linker DNA. To support, and fully evaluate, this claim the following points should be addressed:a) The conclusion from Figure 1 is that nucleosomes are refractory to DNA methylation without chromatin remodelers. However, it is unclear how the correlations between DNA methylation and nucleosomes were conducted. For panels A and C, either the methylation in all the mutants is shown relative to the nucleosome position in WT plants, in which case potential differences in nucleosome position in the mutants should be accounted for, or the methylation is shown relative the nucleosome positioning in each mutant, in which case the sequence differences in the nucleosome cores vs linker regions should be accounted for. Likewise, for the mammalian data, where the methylation data in the WT and lsh mutants is relative to the nucleosome positions in WT samples, changes in nucleosome positions in the lsh mutant should be accounted for in the analysis. Without such information it remains unclear whether the differences in methylation observed in the mutants are associated with changes in nucleosome positions, DNA sequence biases, and/or changes in the ability of DNA methyltransferases to methylate nucleosomal or linker DNA.

We have added supplemental figures and explanatory text in the results to help clarify this issue for Arabidopsis data. We define groups of positioned nucleosomes separately for each genotype in Figure 1 to account for nucleosome position differences between mutants. Our clustering analysis (now shown in Figure 5) shows methylation patterns in all genotypes around the same set of nucleosomes (cluster C2) and reaches the same overall conclusion.

We have now added supplemental data to illustrate that our conclusions do not change when analyzing sets of well-positioned nucleosomes shared by different genotypes (Figure 1—figure supplement 2). This figure also shows that using only WT positioned nucleosomes as methylation anchors masks the depletion of methylation from nucleosomes in ddm1 and h1ddm1 genotypes. This is because only a limited subset of WT well-positioned nucleosomes are also well positioned in the mutants.

Regarding nucleosome positioning in Lsh null cells, we unfortunately do not have Lsh null nucleosome data. We feel that CTCF sites are appropriate for anchoring analysis of methylation in this context; in essence, these can be thought of as ‘shared well-positioned nucleosomes’. Nucleosome positioning at CTCF sites in humans and mouse has been well-described and is generally invariant when averaged across a population of cells (e.g. Fu et al., 2008) as we do in the current work. Furthermore, this subset of CTCF sites is distant from potentially disruptive positioning cues such as gene promoters (see Methods section of Baubec et al. 2015).

Indeed, the CTCF sites we used here are well bound by CTCF regardless of levels of DNA methylation (Baubec et al. 2015, as derived from Stadler et al. 2011) and following the addition of DNMT3a to mutant cells, mCG appears in the presumptive linker DNA of these regions (Baubec et al. 2015). The observation that average DNA methylation in Lsh null cells peaks in the presumptive linker regions at near-WT levels provides strong evidence that the nucleosomes are positioned as in WT. There is also precedent for our analysis: Baubec et al. (2015) show MNase-seq from WT cells (Teif et al. 2012) in relation to methylation data from Dnmt-triple KO mutants.

b) For the lsh mutant, a meta-analysis is presented showing decreases in DNA methylation over well positioned nucleosomes and near WT levels of methylation in the linker regions; thus, methylation is skewed towards linkers. For ddm1/h1ddm1 mutants a similar trend is observed. However, more in-depth analyses should be presented to strengthen the assertion that LSH and DDM1 are acting similarly. For example, is Lsh activity also mainly associated with heterochromatin?

We have added the following sentence to the Introduction section:

“Knockout of mouse Lsh likewise causes major depletion of DNA methylation from repetitive heterochromatin (Dennis, 2001), with more variable effects on methylation in genes and other sequences (Myant et al., 2011; Tao et al., 2011).”

Is the methylation periodicity in mammalian cells disrupted in lsh mutants?

We have added an analysis of Lsh-/- mCG periodicity to Figure 4; indeed Lsh is required for the strong 10 bp periodicity of methylation at the nucleosome.

8) Another major point is that DDM1 enables the methylation of nucleosomal DNA by remodeling the chromatin such that DNA methylation actually occurs on a DNA loop and not directly on nucleosome bound DNA.Here the authors suppose (without any data) that this loop would still retain a bias in DNA methylation resulting in a periodic pattern matching previous in vivo findings. To test their model, the periodicity of DNA methylation was then assessed in a ddm1 mutant background. However, as this mutant causes global decreases in DNA methylation, disruption of the periodic behavior is hard to assess. Furthermore, the decrease in periodicity could either be because the DNA loops aren't created so methylation is blocked or because DDM1 serves a different function that allows MET1 to directly methylate nucleosomal DNA. Thus, these in vivo experiments remain inconclusive. To support their conclusions, in vitro remodeling assays showing that DDM1 activity creates accessible DNA loops that can be methylated by MET1 would be highly informative, although such assays are clearly not trivial. In the absence of such assays, the authors should note the alternative interpretations noted above.

We have separated the model (now Figure 6) and the analysis (Figure 4) and acknowledged the limitations of our methodology.

9) A final point of the paper is that the decoupling of chromatin remodeling and DNA methylation could explain some of the differences in global methylation patterns observed between species.Here the authors find that a subset of well positioned, heterochromatin-localized nucleosomes (cluster C2) have strong linker methylation and low nucleosome methylation in the h1ddm1 background. Based on the similarity of this DNA methylation pattern with those observed in algae, it is proposed that decoupling methylation and remodeling is a mechanism employed to generate such patterns in organisms with well positioned nucleosomes. However, this model is not further vetted. In this case, a phylogenetic analysis correlating the presence or absence of DDM1 or H1 orthologs with linker-specific methylation patterns might provide additional support for the authors claims.

We have added a phylogenetic analysis of histone H1 to illustrate that, as the reviewer suggested, distribution of canonical linker histone may be informative (Figure 5—figure supplement 1). Indeed, canonical H1 is generally absent from the organisms exhibiting linker-specific methylation. No such phylogenetic pattern emerges when analyzing DDM1/Lsh homologs, potentially because these enzymes are multi-functional. Also, species with linker-specific methylation have little heterochromatin and the methylation occurs largely in genes, where DDM1 may have little effect. For these reasons, one would not expect DDM1 loss to strongly correlate with linker-specific methylation.

Although in general we appreciate brevity, some elaboration here would be useful to the reader. Do you think this phenomenon is a result of genetic drift in which in some organisms the loss of one component is not selected against or might it provide some selective advantage?

We suspect that the loss of H1 and gain of linker specific methylation may confer a selective advantage. We have added the following passage to the Discussion:

“The linker-specific methylation patterns almost certainly offer a selective advantage to the species that have them, in part by contributing to nucleosome positioning (Huff and Zilberman, 2014). Thus, loss of H1 could be advantageous in the presence of a linker-specific methylation system, and reliable nucleosome positioning a consequence of such a system.”

In this section is the sentence: "These species are interspersed on the tree of life with species that do not limit methylation to linkers (Figure 4—figure supplement 1)." Figure 4—figure supplement 1 (does this mean Figure 4—figure supplement 1?) does not seem to have much relation to the sentence that cites them – i.e., nothing to do with the phylogenetically "interspersed" nature of the phenomenon because both figures are about Arabidopsis.

This tree may have been improperly referenced in the original submission. Figure 5—figure supplement 2 now contains the cartoon phylogenetic tree depicting the distribution of taxa with linker-specific DNA methylation.

Specific comments:1) Main text, first paragraph: Replace Jones and Baylin, 2007 with Baylin and Jones, CSH Persp. Biol. 2016, which is more appropriate and recent.

We switched this reference as suggested.

2) Main text, first paragraph: To be consistent with the first part of the sentence, I would be more general here and provide a review instead of Ong-Abdullah et al., 2015.

We have added a 2017 review paper by Springer and Schmitz.

3) Main text, fourth paragraph: It is not clear if the four groups of nucleosomes were determined for each of the four genotypes or solely for the WT and then maintained for the analysis of the three mutant genotypes? I suspect that it is the latter, but the description in Materials and methods suggests otherwise. How can meaningful comparisons be made if the four groups of nucleosomes differ for each genotype?

Please see general comment 7a and response above.

4) The authors could easily complement their analysis of heterochromatic nucleosomes by considering separately those that have CMT2-dependent CHH methylation and those that are targeted by DRM1/2 instead. This additional analysis could also prove very informative.

An analysis of differentially-methylated CHH regions has been added as a new main figure, Figure 2. We have added a subsection of the Results to describe this analysis, which is consistent with the overall conclusions of our paper and shows that DDM1 is generally most important in heterochromatic regions.

5) Main text, fourth paragraph: Make also reference to the Materials and methods section ("Description of Arabidopsis genome features").

We have added a reference to Arabidopsis genome features (subsection “DDM1 enables CG and non-CG methylation of DNA wrapped in nucleosomes”, first paragraph).

6) Main text, seventh paragraph: “Figure 2—figure supplement 1” should be Figure 1—figure supplement 2.

This has been corrected.

7) Main text, ninth paragraph: Are all genes non TE-genes? And what about TEs? Please provide about how the TE class was defined and how it differs from the heterochromatic nucleosomes analyzed in the first part.

Clarifying notes have been added to the Results and Materials and methods where appropriate. In particular, our gene annotations exclude TEs.

8) Main text, tenth paragraph: The fact that gene methylation is enriched over nucleosomes and that DDM1 is not required for this type of methylation except in the case of lowly expressed genes suggests that other chromatin remodelers are involved. However, at this stage this is only a hypothesis and the authors should therefore tone down the last sentence of this paragraph.

We have edited this claim:

“Our results demonstrate that DDM1 facilitates methylation of nucleosome-wrapped DNA in a subset of genes as well as in heterochromatin.”

We also now describe the possibility of other remodelers facilitating genic methylation in the discussion, to make clear that this is a hypothesis.

9) Materials and methods:- Why was the mutant allele ddm1-10 chosen over the much more commonly studied allele ddm1-2, used for instance in Zemach et al., 2013.

To prevent complications of using inbred ddm1 alleles that have existed as homozygous mutants and retain methylation defects even when backcrossed to WT, we used this allele, which we obtained in a heterozygous state. We have added an explanation to the Materials and methods section to convey this.

- Which generation was used for ddm1 and h1ddm1? This is important to know as the demethylation of heterochromatic sequences increases with the number of generations.

F4. The reason for is that – as explained in forthcoming work related to this manuscript – we have found h1ddm1 CG methylation at TEs decreases from F2 to F3, whereas by the F4 methylation is stable for the most part.

- Were DNA, RNA and nucleosomes extracted from the same starting biological material?

DNA for bisulfite sequencing was taken from the same tissue type and the same generation as the MNase samples (F4), but not from exactly the same biological material. RNA was from the WT F3 from the parental h1 x ddm1/+ cross. The RNA-seq data were used only to define gene expression deciles.

10) Figure 1 and Figure 1—figure supplement 2: I would recommend that the supplement be moved to Figure 1, as it is clear that CHH methylation is also affected in the three mutant backgrounds.

We have added CHH methylation analysis to the main figure.

I would also recommend for each of the three sequence contexts merging the four graphs, so as to make more apparent the gains and losses of methylation. This is particularly important for CHH methylation, which is low to start with compared to CHG and especially CG methylation and which show modest fluctuations in absolute values. Indicating levels relative to WT could improve visualization further.

We agree that showing methylation around nucleosomes relative to WT helps to convey the phenotype more clearly. We have added a panel, Figure 1, to indicate the relative changes in methylation vs. WT.

11) The legend of Figure 4 is confusing as a result that figure is difficult to understand. The authors should also show mCHG and mCHH data or else justify why they chose not to show them.

We have added heatmaps of methylation for the non-CG contexts to the supplement (Figure 5—figure supplement 1); they show strong periodicity in the C2 cluster, consistent with the metaplots shown in Figure 5.

In Figure 4 it is not clear what data is being shown in the right panels.

This is now Figure 5. We have updated the figure to clarify that this is the same plot as on the left, except zoomed in on the y-axis for ease of seeing that WT and h1 methylation is also phased with nucleosomes. This serves to highlight that there is a modest degree of periodicity in methylation even when DDM1 is present.

12) It is not clear from the methods if the same stage leaf tissue was used for the plant DNA methylation and nucleosome profiling experiments. As these features can vary during development, this point should be clarified.

We have added a sentence at the end of the first Materials and methods paragraph to clarify this point. For all experiments, rosette leaves of 1-month old plants were used.

[Editors' note: further revisions were requested prior to acceptance, as described below.]

There are a few minor points to address that the reviewers and I think would make your paper even more clear to the eLife readership.1)We suggest a more direct formulation. "DNA methylation at DRM1/2-dependent loci is actually enriched around well-positioned nucleosomes in WT (Figure 2). This enrichment is likely caused by the small size of DRM1/2-dependent TEs and hence the high proportion of non-methylated adjacent sequences over these nucleosomes. Consistent with this interpretation, an analysis anchored to the centers of differentially methylated regions (DMRs; see Methods)..."

The issue here is similar to the point above – we did not present our data clearly and described the results in an overly conservative and confusing way. CG and CHG methylation is actually depleted over nucleosomes very similarly at DRM- and CMT2-dependent loci, which shouldn’t be surprising given that the same methyltransferases (MET1 and CMT3, respectively) are responsible. CHH methylation is also depleted over nucleosomes at DRM-dependent loci, but mutating H1 or DDM1 doesn’t make much difference. We now illustrate this by plotting methylation in Figure 2 as we do in the new Figure 1, and by including a new supplementary figure analogous to the new Figure 1—figure supplement 1 (please see response to point 4). We also changed the associated text as follows:

“*Arabidopsis* TEs can be roughly grouped into heterochromatic and euchromatic elements (Zemach et al., 2013). […] Due to the strong linker enrichment of methylation in *h1ddm1* heterochromatin (Figure 1), even analyses centered on DMRs between *cmt2* and wild-type without regard to nucleosome position produce periodic *h1ddm1* methylation patterns (plots in the right column of Figure 2).”

2) Discussion, fourth paragraph, last sentence: This sentence is ambiguous. Gene body methylation is affected by MET1, with no contribution from the other methyltransferases. The ddm1 mutation affects only a small subset of body methylated genes.

We altered this sentence to:

“Despite their preferential methylation, genic nucleosomes – at least those experiencing low levels of transcription – are refractory to MET1 methyltransferase activity, as evidenced by their decreased methylation in *ddm1* mutants (Figure 3).”

3) Re the statement in the Abstract "We find that DDM1 enables methylation of DNA bound to the nucleosome, thus demonstrating that nucleosome-free DNA is the preferred substrate of eukaryotic methyltransferases in vivo." The first half of the statement is well supported, but the data do not demonstrate that nucleosome-free DNA is the preferred substrate. Rather, the data support this hypothesis. It is up to you how to re-phrase, but an example of re-phrasing would be "We find that DDM1 enables methylation of DNA bound to the nucleosome, consistent with a model in which nucleosome-free DNA is the preferred substrate of eukaryotic methyltransferases in vivo."

We changed this sentence of the Abstract to:

“We find that DDM1 enables methylation of DNA bound to the nucleosome, suggesting that nucleosome-free DNA is the preferred substrate of eukaryotic methyltransferases in vivo.”

4) In several cases, you draw specifically on the effects of h1 on CG methylation (see quoted text below). However, the effects of h1 on CG methylation are minimal. Indeed, when plotted relative to WT levels in Figure 1, the line is flat. However, h1 clearly has effect on nucleosome core methylation levels in the CHG and CHH contexts, and it also enhances the ddm1 mutant. In light of these observations the text should be modified to more accurately reflect the data presented."CG methylation is relatively evenly distributed with respect to nucleosomes in WT plants, and shows a weak nucleosomal depletion in h1 mutants (Figure 1).""Without H1, linker DNA becomes even more accessible, producing the observed slight core-to-linker methylation differential in h1 mutants (Figure 1).”

We thank the reviewers for pointing out that our data presentation in Figure 1 is not clear and should convey the relationship between nucleosomes and methylation more accurately. To avoid the appearance of over-interpretation, we actually described our data too conservatively. CG methylation is depleted over nucleosomes in all genotypes, including h1 and WT. This wasn’t obvious in Figure 1, especially for WT, because the figure presented only one full linker-to-linker oscillation, and because the Y-axis methylation interval was fixed at 0.2 to allow for easier comparison between genotypes. In Figure 1, the nucleosome depletion in h1 plants is even less apparent because WT data (also depleted over the nucleosome) are subtracted from h1 data, and the Y-axis range is four times greater than in Figure 1.

To present our data clearly and accurately, we changed Figure 1 panels A, D and E to display methylation for +/-1 kb from the point of alignment. This presents multiple nucleosomal oscillations, which make the relationship between methylation and nucleosomes in all the genotypes far more obvious. We are also including a new Figure 1—figure supplement 1, which shows the same analysis (+/-150 bp) but allows the Y-axis to adjust to the data to emphasize the depletion over the nucleosome in WT and h1. We changed the associated text as follows:

“CG methylation shows a mild but clear depletion over well-positioned nucleosomes in WT and *h1* mutant plants, and an overt depletion in *ddm1* and *h1ddm1* plants (Figure 1, Figure 1—figure supplement 1, loci with well-positioned nucleosomes on the left). […] In the absence of DDM1 and H1, linker DNA is accessible to methyltransferases, but nucleosome cores are not, producing the robust methylation periodicity in *h1ddm1* plants (Figure 1).”

5) The text now makes clear that the methylation data shown in Figure 1 is relative to the specific nucleosome positions in each genotype. The authors also account for possible affects due to differences in the sequences in the nucleosomes and the linkers through their analysis in Figure 1—figure supplement 2. However, to fully evaluate this data, it would be informative to know how many well positioned nucleosomes there are in total and how many are shared between WT and h1ddm1 or between h1 and h1ddm1. In the response to reviewers it is stated that "only a limited subset of WT well-positioned nucleosomes are also well positioned in the mutants." If this overlap is very small, then it remains unclear if this subset of regions can be used to extrapolate information regarding global effects.

We added the relevant nucleosome numbers to the legend of Figure 1—figure supplement 3:

“Nucleosome numbers: 19122 for WT, 4267 shared by WT and h1ddm1 (22.3% of WT), 22745 for h1ddm1, 6364 shared by h1 and h1ddm1 (28.0% of h1ddm1).”

6) The implications of the following statement are unclear."However, this appearance is almost certainly caused by the small size of DRM1/2-dependent TEs, because an analysis anchored to the centers of differentially methylated regions (DMRs; see Methods) between drm1drm2 mutants and wild-type that occur in regions with very poorly positioned nucleosomes (those outside the four groups defined in Figure 1; see Methods) shows an even sharper peak (plots in the right column of Figure 2)."

Please see our response to comment 1 above.

7) To avoid confusion the word "can" should be added to this sentence of the Introduction: "Methylation also regulates endogenous genes: methylation close to the transcriptional start site can cause gene silencing[…]"

We made the suggested change.